# Nanoscale mechanism of UO$_2$ formation through uranium reduction by magnetite

Zezhen Pan[1], Barbora Bártová[1,2], Thomas LaGrange[3], Sergei M. Butorin [4], Neil C. Hyatt [5], Martin C. Stennett [5], Kristina O. Kvashnina [6,7] & Rizlan Bernier-Latmani [1✉]

Uranium (U) is a ubiquitous element in the Earth's crust at ~2 ppm. In anoxic environments, soluble hexavalent uranium (U(VI)) is reduced and immobilized. The underlying reduction mechanism is unknown but likely of critical importance to explain the geochemical behavior of U. Here, we tackle the mechanism of reduction of U(VI) by the mixed-valence iron oxide, magnetite. Through high-end spectroscopic and microscopic tools, we demonstrate that the reduction proceeds first through surface-associated U(VI) to form pentavalent U, U(V). U(V) persists on the surface of magnetite and is further reduced to tetravalent UO$_2$ as nanocrystals (~1–2 nm) with random orientations inside nanowires. Through nanoparticle re-orientation and coalescence, the nanowires collapse into ordered UO$_2$ nanoclusters. This work provides evidence for a transient U nanowire structure that may have implications for uranium isotope fractionation as well as for the molecular-scale understanding of nuclear waste temporal evolution and the reductive remediation of uranium contamination.

[1] Environmental Microbiology Laboratory, École Polytechnique Fédérale de Lausanne, 1015 Lausanne, Switzerland. [2] Interdisciplinary Center for Electron Microscopy, École Polytechnique Fédérale de Lausanne, 1015 Lausanne, Switzerland. [3] Laboratory for Ultrafast Microscopy and Electron Scattering, École Polytechnique Fédérale de Lausanne, 1015 Lausanne, Switzerland. [4] Molecular and Condensed Matter Physics, Department of Physics and Astronomy, Uppsala University, Box 516SE-751 20 Uppsala, Sweden. [5] University of Sheffield, S10 2TN Sheffield, UK. [6] The Rossendorf Beamline at ESRF – The European Synchrotron, CS40220, 38043 Grenoble, Cedex 9, France. [7] Helmholtz Zentrum Dresden-Rossendorf (HZDR), Institute of Resource Ecology, PO Box 51011901314 Dresden, Germany. ✉email: rizlan.bernier-latmani@epfl.ch

Redox transformations from soluble uranium (U) hexavalent species (U(VI)) to insoluble tetravalent species (U(IV)) largely constrain uranium biogeochemical behavior. This redox reaction occurs in the remediation of soils and sediments where biologically generated minerals may immobilize contaminant U[1–3], as well as in paleo-redox studies where the isotopic signature of uranium reduction may be used to indicate the presence of $O_2$ in the geological record[4–7]. Redox-active minerals, including Fe(II)- or sulfide-bearing minerals such as pyrite, mackinawite, magnetite, green rust, and Fe(II)-containing clays[1,2,8–16] are responsible for U(VI) reduction in ore deposits, anoxic aquifers, and marine sediments. Iron(II)-bearing minerals have gained attention also due to their importance in nuclear waste disposal where steel corrosion products may include magnetite ($Fe_3O_4$)[12,17–19].

The molecular mechanism of U(VI) abiotic reduction by Fe (II)-bearing reducing agents, the electron transfer to U(VI), and the subsequent precipitation of U(IV) oxide have not been fully unraveled. Generally, studies to date have suggested that the reduction process consists of U(VI) adsorption followed by electron transfer by structural, adsorbed, or aqueous Fe(II) to result in the formation of U(IV)[3,18,20]. The reaction pathway and kinetics are controlled mainly by aqueous geochemistry conditions[12,21,22] and mineral characteristics, such as the availability of Fe(II) in either solid structures or aqueous phase[1,10,11,18]. Most laboratory studies report the final reduced product as U(IV), occurring as nanoparticulate uraninite ($UO_2$)[3,9–12,18]. Meanwhile, others have observed the formation of non-uraninite U(IV)[8,22], or monomeric U(IV) species due to the presence of ligands or biomass that preclude the precipitation of $UO_2$[1,23]. Moreover, contradictory morphologies have been suggested, including the formation of a coating of $UO_2$ on the surface of mackinawite[2,23], or individual $UO_2$ nanoparticles associated with the edge of green rust particles[9] and large magnetite crystals[12], as well as stand-alone aggregates away from the magnetite surface[11,22]. Thus, a molecular-scale view of the process of formation of uraninite and its crystal growth process is still lacking.

The presence of pentavalent U (U(V)), as an intermediate valence state, has been demonstrated in laboratory experiments[24–26], and its importance as a long-lasting intermediate in the reductive process is starting to be recognized. However, uncertainty about its presence and of its role in reduction pathways involving iron oxides remains. The reduction of U(VI) to U (IV) can occur via (a) two single-electron transfer steps, from U (VI) to U(V), and U(V) to U(IV), or (b) from U(VI) to U(V) followed by disproportionation of two U(V) to U(VI) and U(IV)[22,27–29]. Theoretical calculations reported the reduction from U(VI) to U(V) by aqueous Fe(II) to be facile[29] and demonstrated that the incorporation of U in solid phases widens the stability field of U(V) species in the reduction by magnetite[16]. U(V) incorporation into iron oxide phases has been described experimentally in several scenarios. First, through the reduction of U(VI)-incorporated hematite by aqueous Fe(II)[30] or during the Fe(II)-induced transformation of iron oxides, from ferrihydrite to goethite or magnetite[21,31,32]. Second, it is well established that during the coprecipitation of U and magnetite, U(V) is incorporated into the iron oxides and persists for up to a year[33]. Third, the presence of U(V) has been detailed when U(VI) was reduced by pre-formed magnetite at relatively low pH values (<5) at which dissolution and recrystallization of iron oxides occur[16,26,34]. However, under neutral pH conditions, which is the more likely scenario in reducing soils and sediments, a single study showed the presence of $U_3O_8$ (harboring U(V) and U(VI))[35] with X-ray photoelectron spectroscopy (XPS)[19]. In contrast, most other studies did not show direct evidence of U(V) on the pre-formed magnetite surface[11,22]. It should be noted that U(V) incorporated

in iron oxides is considered to hold a uranate(V) structure because the first shell U–O bond distances are all ~2.0–2.1 Å[33,36], as opposed to a uranyl(V) structure, which has short U–O trans-diaxo bonds (1.9 Å) and longer equatorial bonds (2.50 Å)[37].

Thus, while U(VI) reduction by minerals has been studied for decades, the molecular mechanism of formation of uraninite is mostly unknown. The role of U(V), the transfer of electrons to U from the mineral surface, and the formation of $UO_2$ are all poorly constrained. Furthermore, under some conditions, abiotic U(VI) reduction shows an isotope fractionation behavior deviating from equilibrium[38,39]. Depending on the reducing agent and on the chemical conditions, the isotopic fractionation magnitude and direction appear to vary[38,39]. The mechanism by which this purported kinetic isotope fractionation occurs remains unknown. Clearly, a thorough understanding of the underlying mechanism of U(VI) reduction along with the identification of intermediate species formed would allow a better interpretation of the isotope fractionation behavior. Overall, an understanding of the mechanistic underpinnings of U reduction is essential in better constraining predictions and explanations of its occurrence in the fields of nuclear waste disposal, remediation, and paleo-redox reconstructions.

In addition to synchrotron-based techniques and XPS as tools to probe valence states, we introduce an additional tool, electron energy-loss spectroscopy (EELS), which has been used to unravel the 5$f$ occupancy of actinides at the micro- and nanoscale[40–42]. Correlations between $M_4$ and $M_5$, or $N_4$ and $N_5$ edge intensity ratios and the number of 5$f$ electrons have been demonstrated for actinide elements[40,43]. For U species, the branching ratio, defined as the ratio of the $M_5$ edge intensity to the sum of the intensities of the $M_4$ and $M_5$ edges in EELS spectra, increases with electron occupancy of the 5$f$ orbital, that is, the occupancy of 0 for U(VI), 1 for U(V), and 2 for U(IV)[40,44]. This relationship provides the opportunity to spatially determine the U valence state for heterogeneous samples using nanoscale electron spectroscopy in the transmission electron microscope.

In the present study, we select to work with magnetite nanoparticles and examine the reduction mechanism and morphology of uranium species at neutral pH values. U(VI) is observed to be reduced gradually to a U(V)/U(IV) mixture, which remains for an extended period before complete reduction to U(IV). We discover that the reduction starts with the formation of nanocrystals (1–2 nm diameter) on and around magnetite nanoparticles, followed by their self-assembly into nanowires. The nanowires extend and form a network structure which persists for weeks but eventually disappear with only uraninite nanoclusters remaining. The experimental findings for this system help to expand our understanding of iron–uranium redox chemistry and the stability and bioavailability of uranium species in natural and engineered environments. Additionally, by considering the reductive mineralization of U at the near-atomic scale, this work opens the field to further studies of reduction-induced crystallization at the mineral–water interface.

## Results

**Bulk characterization of reduction products.** Upon amendment of U(VI) to the magnetite suspension, rapid U(VI) adsorption was observed as evidenced by the precipitous decrease in aqueous uranium concentration such that only 1.7% of the initial U(VI) remains in the aqueous phase after 10 min (Supplementary Fig. 1). We applied an extraction with 100 mM bicarbonate, which complexes adsorbed U(VI) surface species while it does not target U(IV)[45]. The extraction suggests that reduction was much slower than adsorption, as exemplified by the slow decrease in the amount of U extracted with the bicarbonate solution (Supplementary Fig. 1). The U $M_4$ edge high-energy-resolution

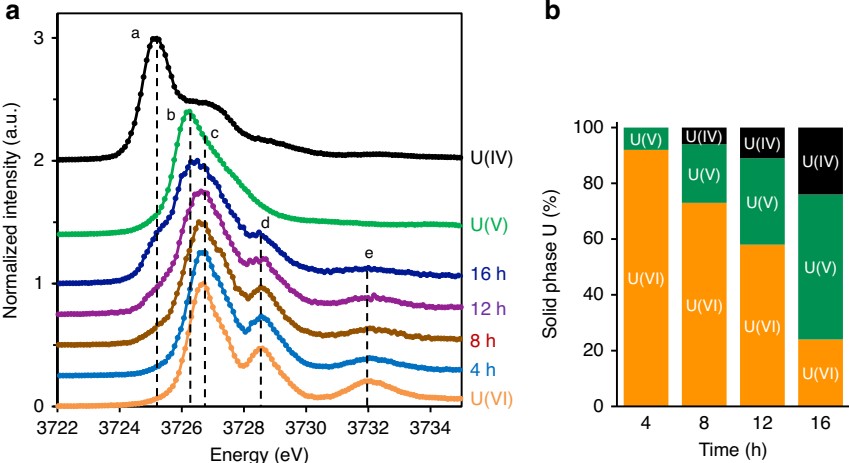

**Fig. 1 X-ray absorption spectroscopy of U-magnetite solid phase. a** $M_4$ edge high-energy-resolution fluorescence detection X-ray absorption near-edge structure spectroscopy ($M_4$ edge HERFD-XANES) spectra for U(VI) reacted with magnetite after 4, 8, 12, and 16 h, comparing to U(IV) (UO$_2$), U(V) (uranate in UMoO$_5$), and U(VI) (uranyl(VI)-nitrate) reference spectra. Dashed lines in **a–c** indicate the white line energy positions for U(IV), U(V), and U(VI) valence states, respectively; **d** and **e** indicate post-edge shoulders for the uranyl(VI) structure. **b** The fractions of U(IV), U(V), and U(VI) components in each U-magnetite solid sample as a function of time as calculated with the iterative transformation factor analysis (ITFA) method. Source data are provided as a Source Data file.

fluorescence detection X-ray absorption near-edge structure spectroscopy (HERFD-XANES) spectra (collected with an X-ray emission spectrometer[46]) obtained from solid-phase samples collected at 4, 8, 12, and 16 h show a shift in the energy position of the absorption edge with time (Fig. 1a), indicating the progressive reduction of U. Furthermore, the two post-edge features characteristic of uranyl(VI) (at 3728.6 and 3732.1 eV), gradually decrease in intensity over time (Fig. 1a). Concomitantly, a shoulder appears in the sample spectra, corresponding to the U(IV) white line position (at 3,725.2 eV, dashed line a), suggesting the reduction of U(VI) to U(IV). This shoulder feature in the spectra is consistent with the U $L_3$ edge extended X-ray absorption fine structure (EXAFS) spectroscopy measurement showing the formation of U(IV) crystalline species over time (Supplementary Fig. 2). More importantly, because of its exquisite energy resolution, the $M_4$ edge HERFD-XANES technique makes it possible to identify the U(V) valence in a mixture of uranium valence states[33,35,36]. Spectra of samples acquired with $M_4$ edge HERFD-XANES were interpreted by iterative-target transformation factor analysis (ITFA) to quantify the contribution of the three different U valence states (Supplementary Fig. 3)[47]. The result suggests the initial reduction of U(VI) to U(V), the appearance of U(IV) at 8 h and the persistence of a U(VI)/U(V)/U(IV) mixture until 16 h (Fig. 1b). The U(V) component increases from 8% to 52% within 16 h, while U(IV) increases from 0 to 24% in the same period. Our observations with $M_4$ edge HERFD-XANES reveal the clear presence of U(V) as a dominant reduction product and evidence of the persistence of U(V) species under neutral pH conditions (Fig. 1b).

We also applied $L_3$ edge HERFD-XANES to confirm the reduction from the mixed-valence (U(VI)/U(V)/U(IV)) to U(IV) for samples equilibrated longer than 24 h. Considering the first derivative of each HERFD-XANES spectrum, we observed a shift of the position of the inflection point to progressively lower energies for 24, 48, 96 h, and 4 weeks of equilibration (Supplementary Fig. 4b). The 4-week sample has the same energy position as a non-crystalline U(IV) sample. Thus, we conclude that the $L_3$ edge EXAFS and $L_3$ edge HERFD-XANES measurements are consistent with the $M_4$ edge XANES results, and both indicate slow reduction from U(VI) to U(IV) via the formation of an intermediate oxidation state (based on the $M_4$ edge data).

The dissolution and recrystallization of iron oxides at lower pH conditions or Fe(II)-induced U(VI) reduction during the iron oxide transformation process can result in the incorporation of U(V) species into the near-surface solid structure, as uranate(V) species[21,26,31,34]. The $L_3$ edge HERFD-XANES spectrum of a U(V) reference consisting of uranate(V) incorporated into the magnetite structure exhibits a broad feature in the white line region (Supplementary Fig. 4), which was also observed in previous studies[33,34,36,48]. This feature is absent in the time course samples, indicating that U(V) detected here is not likely to be incorporated inside the crystal structure of magnetite. In the $M_4$ edge HERFD-XANES measurement[49], uranyl(V) species usually exhibit two post-edge features 1.4 and 3.6 eV from the white line peak (indicated by dashed line b), which represent the short trans-dioxo and equatorial U–O bonds. The $L_3$ edge XANES spectra of uranyl(V)-carbonate complex also exhibits two post-edge peaks[37]. These small peaks were observed neither in the $M_4$ nor in the $L_3$ edge XANES spectra of any of the samples considered, suggesting that uranyl(V) is not the dominant U(V) species. Thus, combining results from the $M_4$ edge and the $L_3$ edge HERFD-XANES, we conclude that U(V) is neither incorporated into the magnetite structure, nor is it a surface uranyl(V). We hypothesize that the species represents a non-uranyl(V) surface species but determining the exact speciation is challenging, and beyond the scope of this work.

**Nanoscale morphology and mineralogy.** To monitor the morphology of reduced U species and identify the localization of U(V) species, we imaged the reduction process at the nanoscale by Scanning Transmission Electron Microscopy (STEM). Additionally, we probed the mineralogy of the U-bearing precipitates by applying fast Fourier transform (FFT) analysis to atomically resolved High-Angle Annular Dark-field STEM (HAADF-STEM) images from samples incubated for 4, 24, 72 h, 5 days, and 4 weeks, and selected-area electron diffraction (SAED) at 72 h. We used the magnetite phase as an internal calibration for precise lattice parameter determination (details of the FFT and SAED data analysis are described in "Method" section and Supplementary Note 1).

After 4 h of reaction, no distinct reduction products were observed. Low- and high-magnification HAADF-STEM images

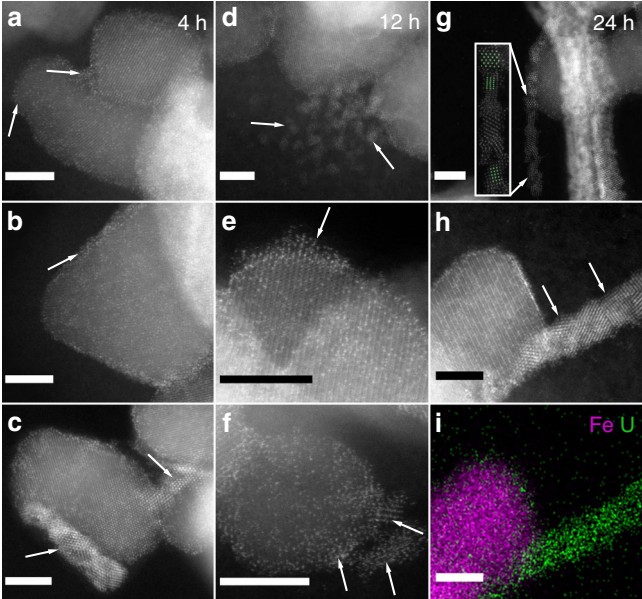

**Fig. 2 Scanning transmission electron micrographs of U-magnetite samples.** High-angle annular dark-field scanning transmission electron microscopy images of U-magnetite samples were obtained (scale bar 5 nm). **a**, **b** 4-h Sample; **c**: replicate 4-h sample; **d–f** 12-h sample; **g–i** 24-h sample. **i** Energy-dispersive X-ray spectroscopy showing U in green and Fe in magenta. The bright contrast spots on the magnetite surface in **a–h** are U atoms. Arrows in **a** and **b** point to dispersed U atoms on the surface of magnetite nanoparticles or accumulated U atoms on the edge of magnetite nanoparticles; arrows in **c** point to formed U nanoparticles in a replicate 4-h sample (see Fast Fourier Transform in Fig. 3); arrows in **d** and **f** point to formed U nanoparticles on or near the surface of magnetite nanoparticles; the arrow in **e** points to the accumulation of U atoms along planes in the same orientations as magnetite lattice planes; arrows in **h** point to individual nanoparticles in a single nanowire that extends from the magnetite surface; inserted image in **g** simulated UO$_2$ pattern (in green) for the three nanoparticles contained in the observed nanowire structure and the defined zone axes: from top to bottom, nanoparticles are in the [011], [112], and [001] zone axis. The simulation of the given zone axis of U atoms in green was done using Vesta[50].

(Fig. 2a, b) show aggregates of magnetite nanoparticles with uranium atoms readily identifiable as bright contrast spots on the surface and prominent accumulation at the particle edges. This observation is consistent with the M$_4$ edge XANES result, showing that the majority of U (92%) remained as U(VI) after 4 h. Therefore, no significant solid-phase uranium oxide products would be expected. In contrast, observation of the 12-h sample shows individual nanoparticles (about 1–2 nm in diameter) on or near the surface of magnetite nanoparticles (Fig. 2d–f; Supplementary Fig. 5b). Single short nanowires extending from the surface of magnetite were observed (12-h sample in Fig. 2f, 24-h sample in Supplementary Fig. 5c) and were interpreted as the early stages of the self-assembly of the U nanoparticles into nanowires, most likely composed of uranium oxide. In a replicate 4-h sample (replicate 4-h sample), while again no significant number of nanoparticles was found, a few areas where nanoparticles organized into short nanowires were observed, suggesting variability in reaction kinetics and/or surface reactivity of magnetite (Fig. 2c; Supplementary Fig. 5a and Supplementary Table 1). Based on the spectroscopy results, we expected that, at the early growth stages, the uranium oxide nanostructures might contain a mixture of U valence states. However, the FFT results for the short nanowires in the replicate 4-h sample establish that

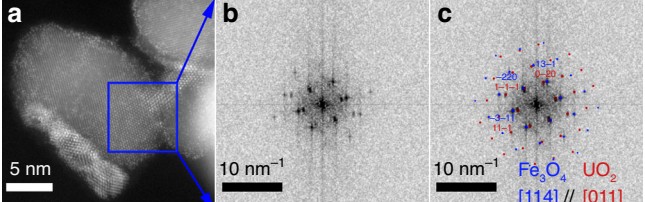

**Fig. 3 Diffraction pattern of a 4-h sample. a** High-angle annular dark-field scanning transmission electron microscopy image for the replicate 4-h sample, showing magnetite particle and uraninite nanoparticle (Fig. 2c). **b** Diffraction pattern in the blue box area in **a**. The blue box area in **a** represents the magnetite grain oriented in the [114] zone axis (blue points in **c**) that is parallel to UO$_2$ nanoparticle in the short nanowire (red points in **c**) which are oriented in the [011] zone axis. Our observation shows that (-11-1) UO$_2$ planes align along specific magnetite planes in this case (2-20), this gives a small lattice mismatch of 6%.

only a UO$_2$ phase (fcc crystal structure, space group 225 and $a = 0.541$ nm) was present in the nanowires (Fig. 3). The diffraction patterns of other possible UO$_{2+x}$ phases, such as U$_3$O$_8$ or U$_4$O$_9$, both of which include U(V), do not fit the experimental data. In those early stages of growth, we also observed that nanoparticles of uranium oxides might exhibit epitaxial growth on the magnetite surface at some locations (Fig. 3). The accumulation of U atoms occurred along planes in the same orientations as lattice planes of the magnetite nanoparticle, exhibiting similar $d$-spacing. This crystallographic correspondence suggests the growth of nanocrystalline uraninite on specific facets of the magnetite about directions that minimize the lattice mismatch (Fig. 2e).

Images obtained from the 24-h (Fig. 2g–i) and especially the 72-h samples (Fig. 4a, b) reveal the presence of bundles of nanowires, which in comparison to the ones observed for the 12-h, show substantial elongation and increase in numbers. The nanowires were confirmed as solid-phase U precipitates by energy-dispersive X-ray spectroscopy EDS (Fig. 2i). Observed nanowires are mostly 5–10 nm wide but can be tens or hundreds of nanometers in length, possibly depending on their stage of formation. More examples of the presence of bundles of nanowires are included in Figs. 5 and 6; Supplementary Fig. 6.

High-magnification imaging revealed that nanowires were composed of strings of nanoparticles, some of which consisted of individual bright spots (e.g., Fig. 4b). The bright spots are columns of U atoms in the uranium oxide nanoparticles, which appear well-oriented about low-index crystallographic directions (Fig. 4b; Supplementary Fig. 5d, e). For the 24-h sample, we identified at least two low-index zone axes as [011] and [001] in a relatively thin and short nanowire by simulation with Vesta[50] (Fig. 2g). While the nanoparticles within the nanowires exhibited average sizes well under 5 nm up until 72 h, at that point in the reaction and beyond, single crystals of larger size (≥5 nm) were identified (Supplementary Fig. 5d, e), suggesting the possible crystallization of uranium oxides. FFT analysis of the nanowire bunches at both 24 (Fig. 5a, b) and 72 h (Figs. 4b and 5c, d) evidenced a ring-like pattern containing spatial frequencies (lattice $d$-spacings) that confirmed the UO$_2$ assignment obtained for the 4-h sample, with lattice $d$-spacings variations within the measurement deviation (Supplementary Table 2). The associated error was within the allowance and compared well with the SAED measurements of the nanowires, which were used to validate the FFT analysis (Supplementary Note 1). The ring-like pattern also supports the presence of multiple orientations of UO$_2$ nanoparticles in nanowires. The 72-h sample was selected for analysis by SAED because of the abundance of nanowires sufficiently distant

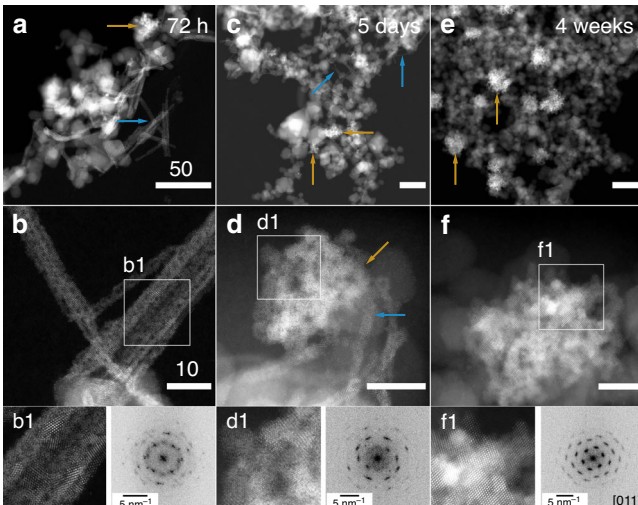

**Fig. 4 Scanning transmission electron micrographs of U-magnetite samples.** High-angle annular dark-field scanning transmission electron microscopy images of U-magnetite samples were obtained (scale bar 50 nm **a**, **c**, **e**; 10 nm **b**, **d**, **f**). **a**, **b** 72-h Sample; **c**, **d** 5-day sample; **e**, **f** 4-week sample. Blue arrows point to $UO_2$ nanowires, whereas orange ones point to $UO_2$ nanoclusters. In **b**, **d**, and **f**, the white squares represent the area that is magnified in panel **b1**, **d1**, and **f1**, where the corresponding Fast Fourier Transform (FFT) was acquired. **b1** Nanocrystals that form nanowires with corresponding FFT showing random orientations. **d1** Nanoclusters formed by similar size nanoparticles with FFT that starts to show texture. The blue and orange arrows point to the nanowires and a connected nanocluster, respectively. This particular morphology suggests that the nanowires collapse into the nanocluster. **f1** Nanoclusters formed by larger nanoparticles, with FFT that reveals all particles have the same orientation (in this case, the [011] zone axis). The streaks in FFT show slight misorientation between the nanoparticles.

from magnetite particles to preclude interference. The complete ring pattern from the SAED measurement further confirms the crystal structure as $UO_2$, as it matched neither $U_3O_8$ nor $UO_3$ (Supplementary Figs. 7, 8 and Supplementary Note 1).

After 72 h of incubation, a second morphology emerges at the expense of the nanowires: it consists of phase-bright nanoclusters covering magnetite particles and becoming the dominant morphology in the 5-day and 4-week incubations. Here, we refer to the bright, contrasted cluster of $UO_2$ nanoparticles in the HAADF-STEM images as phase-bright nanoclusters (pointed out by orange arrows in Fig. 4). At those time points, much less uranium is associated with the magnetite surface relative to the early stages of the reduction, and the morphology of the uranium oxide phases shifts further away from nanowires and more towards nanoclusters. Furthermore, at the 5-day mark, the phase-bright nanoclusters include nanoparticles that are larger and that appear to exhibit more preferred orientations along low-index zone axes (Fig. 4c, d) than those associated with nanowires at 72 h. This trend is evidenced by comparing the FFT pattern for the 72-h nanowires and the 5-day nanoclusters: the 72-h ring pattern suggests the presence of nanoparticles in many different orientations, whereas the 5-day FFT shows streaks. The pattern, between ring and spot pattern, underscores the fact that the nanoparticles start to take preferential orientation at 5 days comparing to 72 h. Furthermore, in the 4-week sample, the observed phase-bright nanoclusters form a streak-spot like pattern, suggesting that a specific orientation of nanoparticles is achieved at this stage. FFT analysis concludes that the observed low-index zone axis (ZA) in Fig. 4f is [011]. Observed nanoparticles in each nanocluster of the 4-week sample have

the same orientation with low-index zone axes (Fig. 5e, f). Small misorientation between nanoparticles was still present in the 4-week sample, resulting in a streak-spot-like pattern instead of a perfect spot pattern. Thus, from 72 h to 4 weeks, the nanoparticles in the nanowires grow, and the nanowires progressively collapse to form nanoclusters of oriented $UO_2$ nanoparticles. The collapse of nanowires into nanoclusters is observable starting at 72 h and for longer incubation periods. At 5 days, nanowires appear to be connected to a nanocluster, suggesting the collapse of the former into the latter (Fig. 4d). Based on the $L_3$ edge HERFD-XANES results, the U species in the 4-week sample is almost entirely U(IV) (Supplementary Fig. 4), and thus, the formation of the $UO_2$ nanoclusters heralds the approach of complete reduction of U(VI) to U(IV). Similar phase-bright nanoclusters were previously reported upon U(VI) reduction by magnetite and were also identified as $UO_2$[22].

Overall, the direct imaging of the lattice structure combined with FFT image analysis at different stages of the reduction (Fig. 5) as well as SAED analysis (Supplementary Figs. 7 and 8) lead to the conclusion that the predominant phase formed in nanowires and in phase-bright nanoclusters is uraninite ($UO_2$) with a U(IV) valence state, ruling out the possibility of mixed U valence states in the nanowires. The appearance of nanoparticles identified as uraninite and their increased abundance corroborate the XANES- and EXAFS-derived conclusions.

An identical-location TEM (IL-TEM) experiment was performed to monitor the reduction process in the same sample over time. This technique allows the repeated probing of specific locations on a grid and thus, the imaging of reaction products as a function of time. Magnetite nanoparticles were deposited on a TEM grid (Fig. 6a), which was immersed into a U(VI) solution to initiate the reduction. After 18 h of incubation, uraninite nanowires were observed at the very location that was imaged before uranium amendment (Fig. 6b). The nanowires formed a network that was morphologically distinct from the magnetite aggregates (Fig. 6c). The TEM grid was immersed once more into the U(VI) solution, and after another 47 h (thus, after a total of 65 h of reaction), the same area was imaged. The nanowires extended further to connect one end of the magnetite agglomerate to the other, forming a bridge (Fig. 6d, e). The rate of nanowire formation and their abundance differed from those observed in batch experiments, likely due to the change in the Fe:U ratio. Nonetheless, through this quasi in situ method, the formation of nanowires and their significant growth during the reduction process was captured, providing strong support for nanowire formation as a result of U reduction by magnetite.

It is intriguing that the specific morphology of uraninite reported here has not been previously observed in the magnetite system, despite a number of studies on the topic. We attribute this discrepancy to the variability in reduction kinetics. Depending on the aqueous composition (pH, carbonate), the reaction conditions (e.g., ratio of U to Fe), and the reactivity of the magnetite surface, the extent and the rate of reduction can vary significantly, varying from 1 day to 10 days of contact time to reach full reduction. A yield of 50 to 100% U(IV) was observed after 1 d of contact time with magnetite in studies with the pH varied between 5 and 10[11,19,34]. In contrast, depending on both the bicarbonate concentration (0 or 2 mM $HCO_3^-$) and the U:Fe ratio, the complete reduction was obtained within 1.6 to 5 days in the study in which similar uraninite clusters were observed previously[22]. In the experiments reported here, the Fe(II)/Fe(III) ratio in the magnetite was determined to be ~0.51, which is the expected stoichiometry. Moreover, the presence of nanowires during the reduction at pH 6.2 and 8 (control 5) also suggests that the formation of nanowires occurs more generally, under varied aqueous chemistry conditions (Supplementary Fig. 6). Overall,

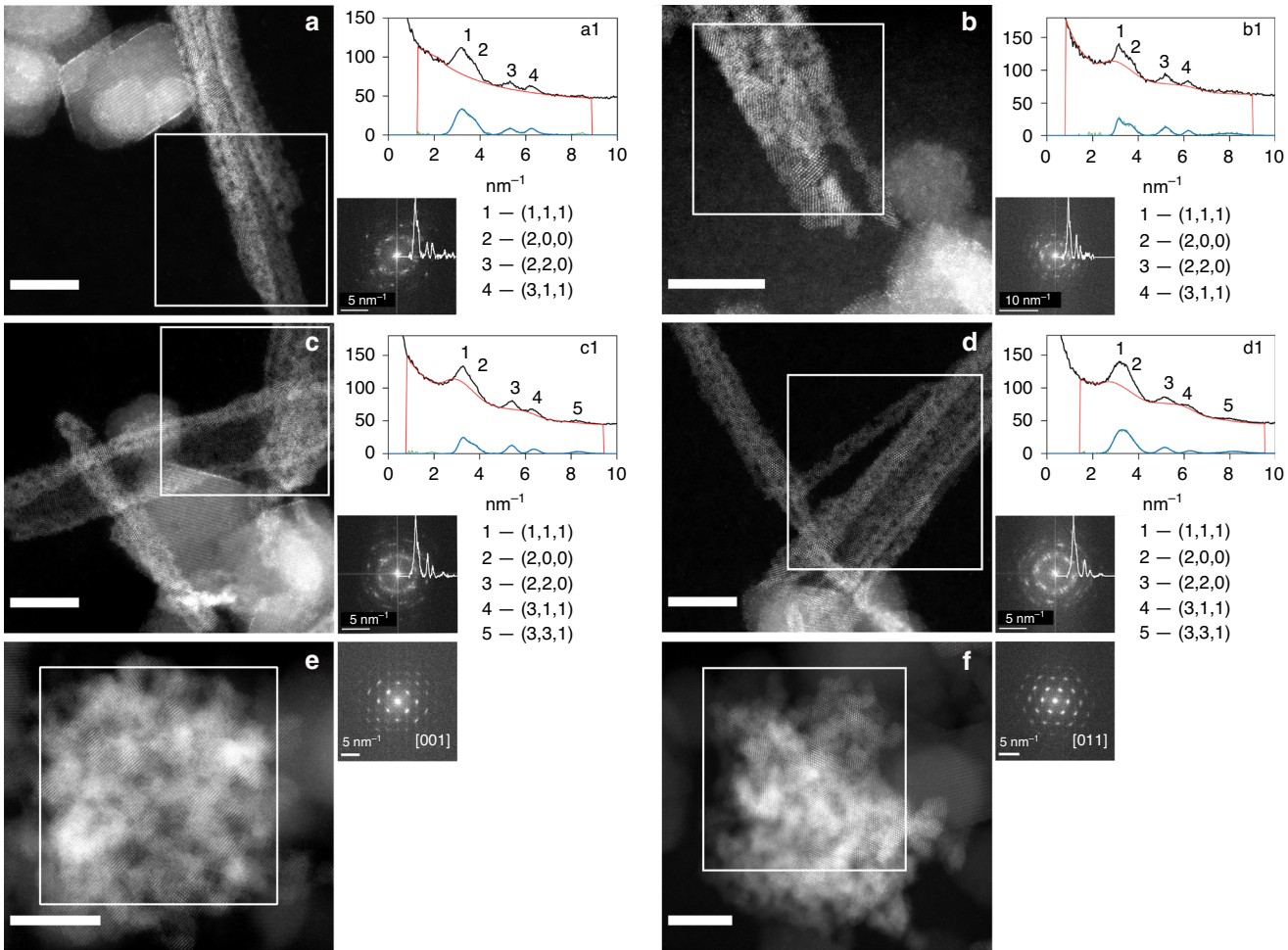

**Fig. 5 Scanning transmission electron micrographs of U-magnetite samples. a, b** 24-h, **c, d** 72-h, and **e, f** 4-week samples and the corresponding FFT acquired from the white square box in each panel (high-angle annular dark-field scanning transmission electron microscopy image scale bar = 10 nm). Panels **a1, b1, c1** and **d1** show radial distribution profiles[73] of the region delineated by white squares in **a–d**, to confirm the assignment to $UO_2$. Black spectra represent the acquired radial distribution profiles; the green profiles are background-subtracted while the blue ones are fitted by Gatan DigitalMicrograph® software (green and blue profiles are very similar and can appear to be indistinguishable). The *d*-spacings obtained from the radial distribution profile were compared to those of a $UO_2$ standard in Supplementary Table 2. For the 4-week sample, **e** and **f** represent well-oriented bright phase nanoclusters that were composed of uraninite nanoparticles in the [001] and [011] zone axes. Panel **d** was also displayed in Fig. 4b.

slow reduction kinetics in the current study were crucial to allow imaging of the transient formation of these structures as well as capturing the intermediate U(V) species. Additionally, the nanowires were readily observable due to the spatial resolution and magnification afforded by Cs-corrected microscopy and to the temporal characterization of the $UO_2$ morphology. In an effort to determine whether such structures could have been observed in previous studies, we scoured seventeen manuscripts considering U(VI) heterogeneous reduction by Fe(II)-containing minerals[1–3,8–15,18,19,22,25,26,34]. Of those, only six used electron microscopy to consider the product of the reduction[1,2,9,14,22,26]. Of those, only one[22] imaged the temporal progression in morphology. However, for that study, the reduction rate was likely faster than in our work, due to a lower U:Fe ratio. In summary, we attribute the novel observation of $UO_2$ nanowires reported here to experimental factors, chief among which are the kinetics of U(VI) reduction and the systematic temporal characterization of the U(IV) product.

Thus, the major contribution of this work is to have captured the intermediate steps in $UO_2$ formation, while reaching the same endpoints as others[22]. The formation of U(IV) nanowires has

been previously reported when glutathione was used to reduce U (VI)[51] as organic molecules could serve as a template for the support/organization of the nanowires in that case. However, in the present system, piperazine-*N,N'*-bis(2-ethanesulfonic) acid (PIPES) was the only organic molecule present, and its role in contributing to the formation of nanowires structure has been ruled out by including a control experiment lacking PIPES (Supplementary Fig. 6c).

**Nanoscale valence state.** We performed spatially resolved measurements of the U valence state in the various U-magnetite samples using TEM EELS (Supplementary Note 2). With a parallel probe of ~200 nm in diameter, we measured the integral intensity of the $M_4$ and $M_5$ edges for uranium and calculated the ratio of their edge intensities, referred to as the branching ratio, which varies according to the valence[44,52]. The experimental parameters and electron dose can influence the branching ratio, and we used uranium oxide standards of known valence to calibrate the branching ratio and to determine the influence of the electron dose. Electron beam-induced reduction was significantly

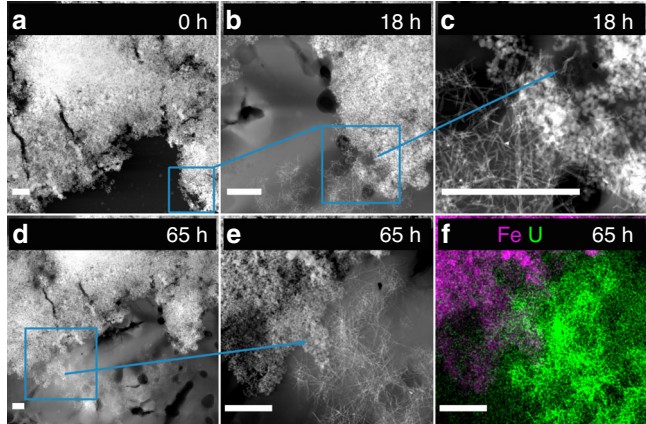

**Fig. 6 Identical-location imaging of magnetite clusters.** High-angle annular dark-field scanning transmission electron microscopy (HAADF-STEM) images of identical magnetite clusters were collected (scale bar 500 nm). **a** At 0 h with only magnetite nanoparticles; **b**, **c** at 18 h showing the formation of U nanowires; the same region is indicated with blue boxes in **a** and **b**; panel **c** represents the blue box in **b**; **d** and **e** at 65 h show further expansion of the nanowires that now bridge the magnetite aggregate (arrow); panel **e** represents the region from the blue box in **d**; panel **f** shows the STEM-EDS map of the region in **e**. The STEM-EDS maps of Fe and U confirm the formation of uranium nanowires after the magnetite cluster reacted in the U solution for 65 h.

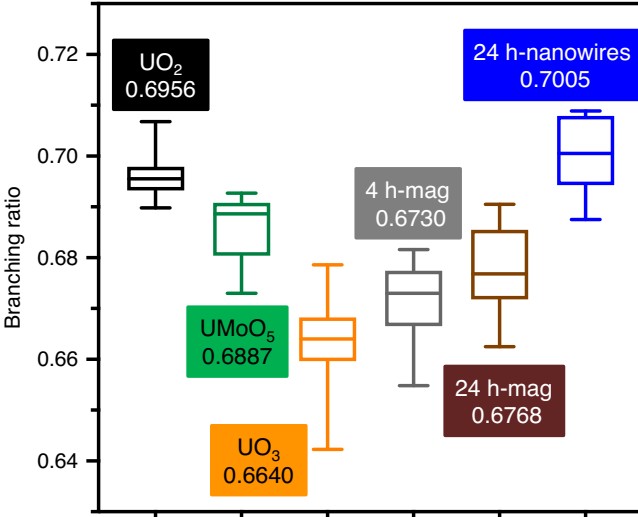

**Fig. 7 Branching ratios from electron energy-loss spectroscopy.** The measurements were obtained in TEM mode. UO₂, UMoO₅, and UO₃ were measured as the reference samples for U(IV), U(V), and U(VI) valence states. The samples considered are: the magnetite surface at various locations for the samples equilibrated for 4 h (4 h-mag) and 24 h (24 h-mag), or U nanowires in the 24-h sample (24 h-nanowires). The results are plotted based on 10–13 measurements for each standard or sample. In the box-and-whisker plot, the box represents the median and the 25th and 75th percentile, while the whiskers extend from the minimum to the maximum value. Source data are provided as a Source Data file.

suppressed at liquid-nitrogen sample temperatures. We observed a decreasing trend of branching ratios for the set of standards, uraninite for U(IV), $UMoO_5$ for U(V) and $UO_3$ for U(VI), from 0.6956, 0.6887 to 0.6640 (median values from 10–13 measurements), providing a reliable reference set to resolve U valence states (Fig. 7; Supplementary Table 3). In a previous study, branching for U(VI) and U(IV) were reported to be

0.633 ± 0.0180 and 0.708 ± 0.011 with mixed-valence states oxides exhibiting ratios in between those values[44]. The smaller range of branching ratios in our study may result from the higher electron doses required to study the lower sample volume of nanowires and to obtain the EELS signal to noise ratio necessary for branching ratio calculation. A detailed discussion of the measurement procedures is provided in Supplementary Note 2.

EELS spectra of U species either associated with nanowires or located on the magnetite surface were measured to obtain the corresponding branching ratios (Supplementary Fig. 9). For the 24-h sample, regions comprised mainly of nanowire bundles were measured, and the obtained branching ratio of 0.7005 agrees well with the U(IV) uraninite reference, confirming that uranium in the nanowires is fully reduced. For regions that mostly comprised adsorbed U complexes on magnetite nanoparticles, the obtained branching ratio was 0.6768. This suggests a valence state between U(VI) and U(IV). There are two possible interpretations of this result. One is that it was challenging to isolate magnetite spatially from the nanowires, and thus the signal reported here could include a U(IV) contribution from the nanowires. The second is that U(V) is the dominant species on the surface, along with contributions from U(VI). To prove our hypothesis that both U (VI) and U(V) are present on the magnetite surface, we also collected EELS spectra on a 4-h sample, which being in the early stages of the reaction, contained no or few nanowires. In this sample, the branching ratio obtained from the magnetite surface was 0.6730, slightly lower than that of the 24-h sample, suggesting a lesser contribution from reduced U species, so a higher fraction of U(VI). This finding supports but does not prove the persistence of U(V) on the magnetite surface.

Fortunately, two more pieces of information are available. The $M_4$ edge HERFD-XANES measurements show the persistence of U(V) at 16 h, and presumably also at 24 h (based on $L_3$ edge HERFD-XANES) and the EELS data for nanowires at 24 h rule out the presence of U(V) in those structures. Thus, the only parsimonious explanation for the persistence of U(V) in the 24-h sample is that it is associated with the magnetite surface.

## Discussion

The combination of bulk and nanoscale techniques in this study provided sufficient information to propose a conceptual mechanistic model that includes valence state and morphological transitions for U(VI) reduction by magnetite (Fig. 8). U(VI) adsorbed on the magnetite surface is reduced to U(V), and progressively to U(IV), which forms dispersed uraninite nanoparticles. The formation of uraninite nanoparticles from soluble species in the aqueous phase cannot be excluded, as the location of uraninite formation could not be directly ascertained from static TEM images. However, based on wet chemistry analyses, the concentration of aqueous U was <1% of the total U after 1 h and 0.5% after 12 h, suggesting that the vast majority of U was associated with the solid phase (through adsorption and reduction). Consequently, the dominant process is heterogeneous reduction, and U(IV) formation most likely occurred at the magnetite surface. We attribute the lack of attachment of individual UO₂ particles to the magnetite surface to electrostatic repulsion between the negatively charged uraninite nanoparticles (pHpzc value close to pH 6[53]) and the either negatively or close to neutrally charged magnetite surface (pHpzc value around 6.9; in the current study at pH values of 7 and 8 (Supplementary Fig. 10)). This repulsion is also expected between the close to neutrally charged uraninite nanoparticles and the slightly positively-charged magnetite surface at a pH value of 6.2, condition at which nanowire formation was also observed (Supplementary Fig. 6e, f). Instead of associating with the magnetite

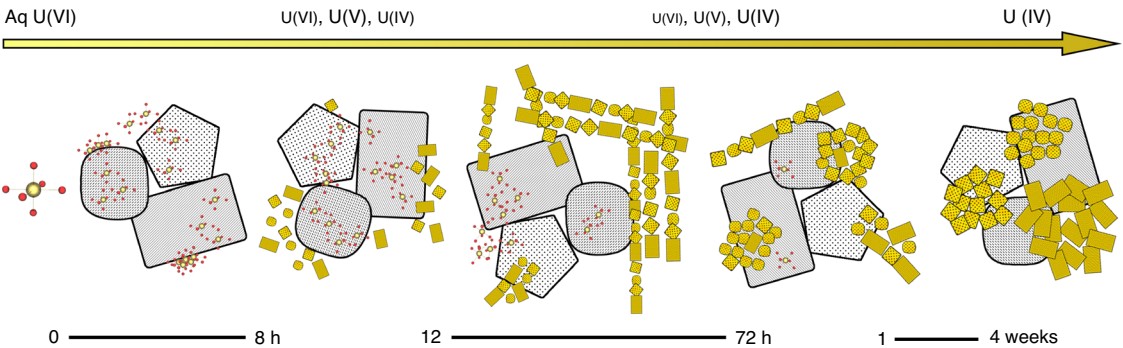

**Fig. 8 Conceptual model of U(VI) reduction by magnetite.** Magnetite particles are indicated in gray and uranium species in yellow. The reaction starts with rapid adsorption of aqueous U(VI) species onto the magnetite surface. Adsorbed U(VI) is reduced to U(V) and U(IV) species. While U(V) remains on the magnetite surface, U(IV) precipitates out as uraninite nanoparticles that self-assemble into nanowires structures anchored to the magnetite surface. Nanowires continue to grow while reduction proceeds. Eventually, crystal growth and coalescence lead to the collapse of nanowires into $UO_2$ nanoclusters in which nanoparticles display a preferred orientation.

surface, nanometer-sized U nanoparticles are attracted by van der Waals forces to reduce the surface energy and self-assemble into nanowires. Previous work with hematite nanoparticles indicates that when nanoparticles are very small, van der Waals attraction force can pull particles/small nanoclusters together to form larger aggregates[54].

The spontaneous self-assembly of U nanoparticles into nanowires that persist for days but ultimately collapse to form phase-bright nanoclusters remains unexplained. The looming questions are, why do nanowires form, and why do they fall apart? The nanowires typically are ~5–10 nm in diameter, ~2–5 uraninite nanoparticles wide, and grow outwards from the magnetite surface. As discussed above, some epitaxial growth of uraninite nanoparticles on the magnetite surface is observable (Fig. 3). We propose that these nanoparticles serve as anchors for the nanowires that grow outwards, away from magnetite, through the attachment of free-floating $UO_2$ nanoparticles repulsed from the magnetite surface by electrostatic forces. Individual nanoparticles in the nanowires exhibit multiple orientations but with predominantly low-index zone axes. Thus, we propose that the nanowire assembly may be the result of crystallization by particle attachment (CPA) where van der Waals forces attract $UO_2$ nanoparticles. Thus, uraninite particles attach to growing nanowires while adsorbed U(VI)/U(V) continues to be reduced to U(IV) oxides and further precipitated uraninite contributes to the growth/extension of nanowires. Individual nanowires connect to others forming bundles and eventually a network of uraninite nanowires.

Oriented attachment (OA), and nearly oriented attachment are potential CPA pathways[55]. Study of the aggregation/attachment behavior of Fe oxide, Ti oxide, and Au nanoparticles during crystal growth has revealed that either OA or aggregation followed by coalescence occurred and resulted in bulk crystals[56–58]. Furthermore, iron oxides in natural samples were shown to form ordered nanoparticle chains via OA, as an example of aggregation-based crystal growth[59]. Such OA requires particle rotation to achieve the correct alignment, as was demonstrated in an in situ TEM experiment of free-floating ferrihydrite nanoparticles[56]. Oriented attachment of nanoparticles has also been invoked in the growth of uraninite crystals during biotic reduction of U(VI) by *Shewanella putrefaciens* CN32[60] and *Desulfosporosinus* spp.[61], though no nanowire morphology was observed. However, the static images in our study revealed random rather than ordered nanoparticle orientations in the nanowires. Perhaps association with a nanowire impedes nanoparticle rotation, hindering attachment-favorable particle-to-particle alignment. A

similar explanation was proposed for the reported alignment about different zone axes of ferrihydrite nanoparticles trapped by other particles[56]. The observation of several low-index zone axes for uraninite nanoparticle may either suggest a nearly OA pattern, or that the particle attachment is followed by coalescence among nanoparticles. Unfortunately, direct evidence for OA or nearly OA is lacking in this study, and future in situ TEM experiments are required to conclusively demonstrate this process.

In the transition from nanowires to phase-bright nanoclusters, the predominance of low-index zone axes remains while the nanoparticles coalesce, forming larger crystals and removing the voids in the nanowire structure. Coalescence between nanoparticles is usually driven by thermodynamics for surface energy reduction, which has been reported to occur for nanoparticles at room temperature[62,63]. During coalescence, re-arrangement occurs on contact surfaces and nanoparticles reorganize into aligned planes to form larger particles with the same orientation. At 5 days, the transition is incomplete as $UO_2$ nanoparticles do not all have the same orientation, but at 4 weeks, the nanoclusters exhibit almost exclusively a single low-zone axis orientation (e.g., [011] ZA) and only uraninite nanoclusters (no nanowires) were observed, heralding the end of the reduction. The nanoclusters occur in association with magnetite (Fig. 4). Detached uraninite nanoclusters have been observed in a previous study where the authors proposed either the formation of $UO_2$ on the mineral surface followed by detachment or the formation of $UO_2$ in solution from aqueous U(V) or U(IV)[22]. With the additional insights brought by this study, we can now conclude the formation of $UO_2$ from surface-associated-U(V) reduction followed by the detachment of uraninite nanoparticles, the formation of nanowires, and their ultimate collapse into $UO_2$ nanoclusters.

The findings reported here of the transient valence state and nanowire structure significantly improve our understanding of molecular-scale mechanisms during U(VI)–magnetite interactions and advance our interpretation of the environmental behavior of U species. This detailed mechanistic understanding of the steps of formation of $UO_2$ at the magnetite surface can serve to refine the fate and transport models of uranium in the subsurface and to relate the isotopic fractionation behavior occurring at mineral–water interfaces to the underlying mechanistic details of reduction. More generally, the uranium-magnetite system represents an ideal model system to delineate the CPA pathway(s) in the heterogeneous reductive mineralization process.

Naturally, more remains to be investigated, particularly regarding the universality of this finding. Will other mineral surfaces, e.g., mackinawite, exhibit a similar mechanism? Will

nanowires also form during biotic U(VI) reduction? This possibility in biotic systems will require cryo-electron microscopy to unravel. Future in situ experiments with systematic observations of the crystallization process are required to decipher the pathways of CPA, e.g., the dynamic migration, orientation, and attachment of nanoparticles, in order to gain insight into crystal growth after nanoparticle formation at the mineral–water interface. Furthermore, the impact of the transient intermediate pentavalent valence state and the nanowire morphology on uranium isotope fractionation and other environmental behavior of U species opens novel avenues for the investigation of uranium accumulation in ore deposits, its mobility in contaminated soil, and its accumulation in ocean and lake sediments. The novelty of this work lies in the nanoscale resolution of the mechanism of U (VI) reduction as well as the identification of a transient valence state and novel morphologies in heterogeneous reductive mineralization[45]. More broadly, the present work may prove to be fertile ground for research into mineralization in reductive environments and the associated role of intermediate species and transient morphologies.

## Methods

**Magnetite synthesis.** All experiments, including the preparation of magnetite nanoparticles, were performed in an anoxic chamber (MBRAUN) with an atmosphere of $N_2$, with $O_2 < 0.1$ ppm. All reagents and chemicals were ACS grade. Aqueous solutions were prepared with milli-Q water (18.2 MΩ cm) and were deoxygenated by purging with $N_2$ before transferring into the anoxic chamber. All glassware and stoppers were cleaned with isotopic grade 6 N HCl, rinsed with water, and dried before any usage.

Magnetite was synthesized based on a precipitation method with a protocol modified from Wang et al.[64]. In brief, anoxic solutions of 1 mol $L^{-1}$ $FeCl_2$ (50 mL) and 1 mol $L^{-1}$ $FeCl_3$ (100 mL) were mixed and continuously stirred inside the chamber. The pH of the mixture was steadily increased by gradually adding 1 M NaOH until the pH value was ~11. The as-synthesized magnetite precipitate was sealed in a serum bottle overnight, then separated by an Nd magnet, and washed with milli-Q water twice. Washed magnetite was resuspended in milli-Q water and sealed in a clean serum bottle with a butyl rubber septum before further usage (always inside the anoxic chamber). Two aliquots (100 μL) of magnetite stock suspension were dissolved outside of the chamber in concentrated $HNO_3$ with a heating plate and then diluted into 1% $HNO_3$ to determine the magnetite stock concentration.

**Colorimetric method.** Magnetite stoichiometry was characterized by its acidic dissolution in 3 M HCl in an anoxic chamber, followed by the measurement of aqueous $Fe^{2+}$ colorimetrically by complexation with the ferrozine reagent (HEPES buffer, pH 7)[65]. Total Fe was measured after reduction of $Fe^{3+}$ by hydroxylamine hydrochloride. Fresh stock solutions of ferrozine reagent, Fe(II) standards and hydroxylamine hydrochloride were prepared the same day as the measurement.

**Uranium reduction experiment.** U reduction experiments were conducted anoxically in triplicates in serum bottles (200 mL). A pH-buffered medium with a pH of 7 was prepared that contained 20 mM piperazine-N,N′-bis(2-ethanesulfonic) acid (PIPES) and 1 mM $NaHCO_3$ and 200 μM uranyl(VI) chloride (from uranyl (VI) chloride in 0.1 N HCl). The mixture was equilibrated for at least 2 h and the initial U concentration was measured before introducing magnetite (final Fe concentration of 5 mM) to initiate the reaction. Serum bottles were kept inside a gray box to exclude photoreduction. Aliquots (1 mL; 10 mL when solid samples were needed for characterization purposes) were collected at time intervals and were placed on an Nd magnet to obtain an aqueous phase and uranium-associated magnetites. The aqueous phase was then filtered through 0.22 μm filters (PTFE, ThermoFisher, USA) to quantify the remaining dissolved uranyl species in the filtrate (aq U(VI)). Meanwhile, aliquots (0.5 mL) were collected in parallel to be mixed with $NaHCO_3$ (at a final concentration of 100 mM) for 30 min, which aimed to desorb unreduced U(VI) or the mobile portion of the reduced uranium species into solution[45]. Again, the mixture went through magnet-separation to obtain an aqueous phase that was then filtered through 0.22-μm filters to obtain another type of filtrate (bicarb-filtrate). Collected filtrates were acidified to 0.1 N $HNO_3$ to preserve them for ICP-MS (Perkin-Elmer) analysis. A control experiment with no magnetite was performed and the U aqueous concentration remained the same as the initial value, suggesting no loss of uranium through the monitored experiment duration. Following the experiment, solid-phase U subsamples were collected for solid characterizations including XAS and TEM techniques, as described in the "Method" section. A full list of the samples and their characterization is included in Supplementary Table 1. For solid characterization, the U(V) reference standard

$UMoO_5$ was synthesized by solid-state reaction of equimolar quantities of $UO_2$ and $MoO_3$. The reagents were intimately mixed using an agate mortar and pestle and sealed in an evacuated quartz ampoule. The mixture was reacted at 900 °C for 24 h and cooled naturally to room temperature[66,67].

Adsorption reduction: U(IV) has low solubility and mobility in bicarbonate-rich solutions, while U(VI) can be readily complexed with bicarbonate and extracted from the solid phase[45]. Bicarbonate extraction (100 mM $NaHCO_3$) was applied to extract the more mobile U species adsorbed on magnetite particles (Supplementary Fig. 1). After contacting them with bicarbonate solutions, the adsorbed U(VI) was released into aqueous phase, and its concentration measured.

**Zeta potential measurement.** Magnetite suspensions (in solutions with 0.02 M ionic strength (NaCl)) were prepared inside the anoxic chamber ($O_2 < 0.1$ ppm), and the pH of each suspension was adjusted by 0.5 M HCl and NaOH solutions. During sonication (for 15 min), each suspension was sealed in serum bottles, which were only opened immediately before zeta potential measurement with a dynamic light scattering device (Zetasizer, Malvern Nano ZS, UK).

**U X-ray absorption spectroscopy.** Uranium valence states were determined in solid bulk phases as a function of time in the reduction experiments using x-ray absorption near-edge structure (XANES) spectroscopy at the U $L_3$ edge (17,166 eV) and $M_4$ edge (3725 eV) with high-energy-resolution fluorescence detection (HERFD). The local atomic coordination of U was determined by extended X-ray absorption fine structure (EXAFS) spectroscopy at $L_3$ edge (17,166 eV). For the U $L_3/M_4$ edge XAS experiments, uranium-associated magnetite solids were separated from the supernatant using an Nd magnet. The supernatant was decanted extensively, and the remaining wet paste was packed for each beamline. The amount of uranium in the remaining pore water of wet paste had an insignificant contribution to the measured spectra.

The sample holders and Kapton films were transferred inside the anoxic chamber for least two days ahead of loading samples to exclude any possibility of reoxidation of the samples. For $L_3$ edge HERFD and $L_3$ edge XAS, wet pastes were mounted and enclosed with Kapton tape and then placed in Nalgene cryovials for measurements. For $M_4$ edge HERFD, samples were loaded onto depressions (1–3 mm thick) within plexiglass plates that were sealed with a layer of 8-μm-thick Kapton film. The second layer of 13-μm-thick Kapton tape was added. Closed or sealed cryovials/plexiglass plates were immediately frozen by placing them inside a liquid-nitrogen cooled cold-well inside the anoxic chamber to minimize any change in the samples. Sealed frozen samples were then stored in the freezer inside the anoxic chamber until transport to beamlines. Each cryovial/plexiglass plate was sealed within a mylar bag before being placed in a hermetically sealed stainless-steel shipping anoxic canister (Schuett-Biotec GmbH, Gottingen, Germany) to be shipped to each beamline. The canister was placed in dry ice during transport. Detailed measurement conditions for each beamline are described in the following section.

$M_4$ edge HERFD-XANES. U $M_4$ edge HERFD-XANES measurements were performed at the European Synchrotron Radiation Facility (ESRF), beamline ID26[68]. The incident energy was selected using the <111> reflection from a double Si crystal monochromator. Rejection of higher harmonics was achieved by three Si mirrors at angles of 3.0, 3.5, and 4.0 mrad relative to the incident beam. The beam size was estimated to be 150 μm vertically and 300 μm horizontally. XANES spectra were measured in HERFD mode using an X-ray emission spectrometer[46,69]. The sample, analyzer crystal, and photon detector (silicon drift diode) were arranged in a vertical Rowland geometry. The U HERFD spectra at the $M_4$ edge were obtained by recording the maximum intensity of the U Mβ emission line (~3337 eV) as a function of the incident energy. The emission energy was selected using the <220> reflection of five spherically bent Si crystal analyzers (with a 1 m bending radius) aligned at a Bragg angle of 75°. The paths of the incident and emitted X-rays passing through the air were minimized to avoid losses in intensity due to absorption. The intensity was normalized to the incident flux. A combined (incident convoluted with emitted X-rays) energy resolution of 0.4 eV was obtained as determined by measuring the full width at half maximum (FWHM) of the elastic peak. The present data are not corrected for self-absorption effects. The analysis shown in this work is based on the comparison of the energy position of the main transitions at the U $M_4$ edge, which is only but insignificantly affected by self-absorption effects. Samples on sample plates (sealed by Kapton film window as described above) were analyzed within a $LN_2$ cryostat. Each spectrum was collected in 2 s, and the beam was moved around the sample to avoid beam damage. Analysis of the spectra was performed by iterative transformation factor analysis (ITFA) to determine the proportion of U(IV), U(V), and U(VI) in samples. The ITFA has been applied to successfully identify three U valence states in $M_4$ edge spectra in previous studies[33,36].

$L_3$ edge HR-XANES and $L_3$ edge XAS. $L_3$ edge HERFD-XANES and $L_3$ edge XAS measurements were performed at the Diamond Light Source, beamline I20-Scanning. Samples were analyzed while contained within sealed cryovials within an $LN_2$ cryostat ($L_3$ edge XAS) or on an $LN_2$ cryojet for $L_3$ edge HERFD-XANES. The beam size was estimated to be ~400 μm vertically and ~300 μm horizontally. A Si (111) 4-bounce monochromator was used for both experiments. For $L_3$ edge XAS, the fluorescence signal was collected with a 64-element Ge detector (Canberra).

The $L_3$ edge HERFD-XANES measurements were also acquired under fluorescence mode but with 3 Ge(111) crystal analyzers and a 4-element Medipix detector. Energy calibration was performed on the first inflection point of yttrium (Y) foil reference (17,038 eV) or based on the energy position of the inflection point of the U(VI) reference sample. To avoid beam damage, each data collection was performed on different spots of the sample. The $L_3$ edge XAS spectra of $UO_2$ were collected at beamline B18 at DLS and corrected based on the energy calibration of reference samples to be comparable with the sample spectra collected at beamline I20 at DLS. The $L_3$ edge HERFD-XANES spectrum of non-crystalline uraninite (serving as the U(IV) standard) were collected at Rossendorf Beamline of the European Synchrotron (ESRF) in Grenoble[70]. The incident energy was selected using the <111> reflection from a double Si crystal monochromator. Rejection of higher harmonics was achieved with two Rh mirrors at an angle of 2.5 mrad relative to the incident beam. XANES spectra were measured in HERFD mode using an X-ray emission spectrometer[46]. The sample, analyzer crystal and photon detector (silicon drift diode) were arranged in a vertical Rowland geometry. The U $L_3$ edge HERFD-XANES spectra were obtained by recording the maximum intensity of the U $L_{\alpha3}$ emission line (13.614 keV) as a function of the incident energy. The emission energy was selected using the <880> reflection of five spherically bent Si crystal analyzers (with 0.5 m bending radius) aligned at 71.5° Bragg angle. The intensity was normalized to the incident flux. A combined (incident convoluted with emitted) energy resolution of 3.2 eV was obtained as determined by measuring the full width at half maximum (FWHM) of the elastic peak. Spectra were corrected based on energy calibration of reference samples to be comparable with the set of data acquired at DLS. Spectra were extracted and processed using the Athena analysis packages[71].

**Transmission electron microscopy**. Synthetic magnetite and uranium associated with magnetite solids were separated from the aqueous phase by an Nd magnet followed by decantation of the liquid. Collected solids were dispersed into 70% ethanol solution and sealed in a serum bottle anoxically and sonicated for 3 min. A drop of the sonicated suspension was then deposited onto an ultra-thin carbon grid (Electron Microscopy Sciences CF200-CU-UL; 200 um square mesh; 3–4 nm carbon foil; copper grid; silicon free) and was immediately transferred into a vacuum desiccator for preservation before the measurement. The sample spent <2 min under ambient conditions. High-angle annular dark-field scanning transmission electron microscopy (HAADF-STEM) images were collected on a double-aberration corrected Titan 60–300 transmission electron microscope (Thermo-Fisher Scientific™) operated at 300 keV with ~80 pA beam current. The same measurement conditions were applied to acquire images for control samples 1 through 4 as described later.

Diffraction pattern analysis. Diffraction pattern analysis on single-crystal nanoparticles as well as orientation relationship simulations was done using JEMS software[72]. Experimentally obtained Fast Fourier Transform (FFT) pattern of selected $UO_2$ and $Fe_3O_4$ nanocrystals (Fig. 3) were indexed based on following crystal files $UO_2$-1541665 and $Fe_3O_4$-9002316 from Crystallography Open database. Other possible structures $U_3O_8$ and $U_4O_9$ were applied but not fitting with the experimental data. Ring FFT pattern from bunches of nanowires was analyzed using the radial distribution profile[73] function in Gatan DigitalMicrograph® software. The radial distribution profile was plotted by a graphical program, and peaks were measured and compared with studied structures (Fig. 5).

Identical-location TEM. A magnetite nanoparticle suspension (5 mM as Fe, equilibrated with 20 mM PIPES and 1 mM $NaHCO_3$) was drop-casted on a customized chip patterned with a 50-nm-thick $SiN_x$-membrane electron-transparent window. The magnetite-deposited chip was then mounted into a holder inside an anoxic chamber and the system was transferred to a ThermoFisher Scientific Tecnai Osiris transmission electron microscope for imaging. High-angle annular dark-field STEM (HAADF-STEM) images of different magnetite clusters were acquired at 200 kV to observed the evolution in their structure. The same chip was retrieved and immersed into a U-containing solution (200 μM, equilibrated with 20 mM PIPES and 1 mM $NaHCO_3$) to undergo reaction for 18 h. After 18 h, the chip was retrieved and placed in the microscope to acquire HAADF-STEM images of the same magnetite cluster. The same procedure was followed for the 65-h time point.

All manipulations were performed in an anoxic chamber ($O_2 < 0.1$ ppm), and the membrane chip was sealed anoxically to be transferred between the anoxic chamber and the microscope. The loading procedure was kept short to minimize exposure of the chip to $O_2$.

Sample preparation conditions for TEM control experiments. Several control samples were prepared to exclude the contribution of artifacts to the formation of nanowires. We considered the impact of using a magnet to separate magnetite from the suspension (control 1), the use of ethanol to disperse the sample before imaging (control 2), and the use of a PIPES buffer for the experiments (control 3). Additionally, we synthesized a separate batch of magnetite to ensure that the observations were reproducible (control 4). Two batches of adsorption-reduction experiments were performed at pH 6.2 and 8 to investigate the extent of nanowire formation at varied pH conditions (control 5). All samples were imaged under the same conditions with double-aberration corrected Titan 60–300 as the samples presented elsewhere in this paper.

Control sample 1 (Contr 1): to rule out the impact of using an Nd magnet during the separation of U-associated magnetite from the suspension, the solid phase was collected as follows: 200 μL suspension was placed in an Eppendorf tube to enable nanoparticle settling. After removing the supernatant, the remaining solid phase was dispersed into 70% ethanol.

Control sample 2 (Contr 2): U-associated magnetite solids were dispersed into DI water to examine the impact of the solvent used for dispersion.

Control sample 3 (Contr 3): to exclude the possibility that the formation of nanowires might be due to the presence of PIPES buffer, a control experiment with no PIPES was performed, with the same loading of magnetite, uranyl chloride, and bicarbonate concentrations. The pH of the mixture was adjusted by the addition of 0.1 N HCl and 0.1 N NaOH.

Control sample 4 (Contr 4): a new batch of magnetite stock was produced to probe whether the formation of the nanowire structures was independent of the magnetite batch.

Control sample 5 (Contr 5): to investigate whether the formation of nanowires is related to a specific pH condition, a control experiment was performed with the pH value adjusted to 6.2 or 8, while maintaining the same magnetite loading, uranyl chloride, and bicarbonate concentrations. The pH of PIPES buffer was adjusted to control the pH of the reaction solutions

Additional methods are available in the Supplementary Information.

## Data availability

All relevant data and images are available from a data repository as https://doi.org/10.5281/zenodo.390456[74] or on request from the authors. Source Data file includes all X-ray adsorption spectroscopy measurements, acquired branching ratios from EELS spectra, reduction kinetics, zeta potentials measurements, and selected-area electron diffraction profile. All original scanning transmission electron microscope images and electron energy-loss spectroscopy data are included in the data repository. Source data are provided with this paper.

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

## Acknowledgements

We thank Luca Loreggian at École Polytechnique Fédérale de Lausanne (EPFL) for helpful instructions and discussions on the project and also his contribution in acquiring reference spectra and also Margaux Molinas for assistance with Identical-location Transmission Electron Microscopy (IL-TEM). We thank Vasiliki Tileli and Tzu-Hsien Shen at EPFL for their instructions and help in performing IL-TEM experiment. We thank Duncan T.L. Alexander and Emad Oveisi at EPFL for helpful microscopy discussions. We thank Daniel Giammar and Anshuman Satpathy at Washington University in St. Louis for providing a UO$_2$ reference sample. The work at EPFL was supported by Swiss National Science Foundation Grant 200021E-164209 and European Research Council Consolidator Grant 725675 (UNEARTH). The M$_4$ edge high-energy-resolution fluorescence detection X-ray absorption near-edge structure spectroscopy (HERFD-XANES) was carried out at Beamline ID26, European Synchrotron Radiation Facility. The L$_3$ edge HERFD-XANES and L$_3$ edge extended X-ray absorption spectroscopy fine structure were carried out at Beamline I20, B18, DLS, and Beamline BM20, ESRF. We acknowledge Diamond Light Source (DLS) for time on Beamline I20-Scanning under proposals SP20581-1, SP20581-2, and Beamline B18 under proposal SP15955. We thank beamline scientists Fred Mosselmans and Shu Hayama for beamtime assistance. The research leading to this result has been supported by the project CALIPSOplus under Grant Agreement 730872 from the EU Framework Programme for Research and Innovation HORIZON 2020. The work at ESRF was supported by European Research Grant 759696 (TOP). S.M.B. acknowledges support from the Swedish Research Council (research grant 2017-06465).

## Author contributions

R.B.-L. and T.L. designed research; Z.P., B.B., and T.L. performed research: T.L., S.M.B., N.C.H., M.C.S., and K.O.K. contributed new reagents/analytic tools; Z.P., B.B., T.L., K.O.K., and R.B.-L. analyzed data; and Z.P., B.B., T.L., and R.B.-L. wrote the paper.

## Competing interests

The authors declare no competing interests.
