## [Peer Review File · Nature Communications]

Reviewers' comments:

Reviewer #1 (Remarks to the Author):

In the present manuscript Pan and co-workers describe a new twist to an old story – reduction of U(VI) by the Fe(II,III) oxide magnetite. The new twists they put to the story are that they (i) have resolved the kinetics of the reduction of U(VI) through U(V) to U(IV) using high-resolution XAS data, and (ii) used detailed high-resolution TEM imaging and spectroscopy to further track the reduction and U(V)/U(IV) transformation process. The observation of this phased transformation from U(VI) to surface U(V) and UO₂ nanowires then to UO₂ clusters with progressively more order is very fascinating and certainly of interest to the community that studies molecular-scale mineral transformations. Detailing this step-wise U transformation process is the main finding of the work. Their findings are on par with the interesting transformations observed for Fe and Ti oxides like oriented attachment-driven crystal growth and fits well with those bodies of work. I also wonder if their data might bring into question some of the mechanisms of bacterial U reduction as well.

I was excited to review the paper when I read the abstract about the UO₂ nanowires transforming to clusters. I was always curious to know why U(IV) had formed clusters in our U(VI)/magnetite experiments at high U:magnetite loadings. The work here is the beginning of an answer to that question, along with some more interesting observations along the way.

The paper is very well written. It has to be in the top 2% of all papers I've ever reviewed for putting together the data into a concise and well-argued manuscript. Kudos to the authors on their hard work! My recommendation is to accept the article with minor revisions. The most significant of these minor revisions would be to assess all alternative hypothesis that might explain the observations and rule them out if they can be with the data at hand – particularly the role of any solution species in the formation of the nanowires or clusters. I think an expanded section ca. line 443 detailing alternative hypotheses and why they can or cannot be rejected would be an excellent addition (or how they might need to be tested further). Some thoughts on what those alternative hypotheses could be are described below along with other specific comments.

Other specific comments/questions:

1. So that others might repeat your experiments: what is the Fe(II)/Fe(III) stoichiometry of the magnetite used in this study? Do you have any data or left-over anoxic samples to point towards this? (But totally understood if not.) Our experience has been that washing results in variable non-stoichiometric magnetite Fe(II)/Fe(III) contents. It is likely that a moderate Fe(II)/Fe(III) ratio is the key to the slow U(VI)/U(V) reduction kinetics.
2. Are there any comparable systems in the other oxides (Fe, Ti, Si(?)) or metals (Pt, etc.) that show similar behavior as suggested here?
3. Why is the assembly of the UO₂ nanowires not the (for lack of a better word) "clicking-in" type reaction observed for Fe oxides (e.g. Li et al, Science, 2012 DOI: 10.1126/science.1219643), where particles move through Brownian motion until they find a preferred low energy direction and stop and "heal" into the underlying crystal?
4. In the paragraph starting on Line 443: It is clear that the data do point towards the fact that some UO₂ nanoparticles are assembled on the surface of the magnetite. However, it is also possible that either exclusively or concurrently assemblage of the "planktonic" UO₂ particles occurs through a solution species (either U(V) or U(IV)). There is not data to rule this out, and it might be a more careful approach to include alternative hypotheses for the data in this discussion.
5. Along the lines of comment 4: It is less plausible to me that the assemblage of the extensive nanowire networks shown in Figure 6 occurs via the surface templating reaction over a distance of several microns. What does the rate of this reaction have to be for this to happen over 65 h? Is it

plausible for the surface-templated and assembly reaction to occur over this time frame? (I don't have a good answer on whether this calculation is possible..)

6. What is the solution U concentration with time in absolute concentration moles/L or ng/L? ICP-MS was used, so likely you should be able to detect and quantify down to the solubility limit of UO₂.

7. Is UO₂ conductive enough to transport electrons from magnetite to any U(VI) or U(V) at the distal end of UO₂ nanowires? E.g. is this nanowire growth an electrochemical reaction occurring at the UO₂? The nanowires remind me in a way of the dendrites you might observe for the reduction of a coinage metal at an electrode (Ag, Cu... etc.).

8. What is the aqueous Fe(II) concentration? Could aqueous Fe(II) be involved in the observed behavior through aqueous U(V)/U(VI) reduction?

9. Why was a uranyl(V) compound not included in the analysis? I am not fully convinced that the surface U(V) is uranate(V) rather than uranyl(V). Ikeda and Hennig have published some LIII XANES and EXAFS data and the post-edge data in Figure S4 seem to fit more with the uranyl(V) shape than the post-edge of the uranate(V) compound shown (Ikeda et al 2007 Inorg. Chem. 10.1021/ic070051y). But I had trouble overlaying their data with yours crudely in powerpoint due to lower white line intensity in Ikeda, so a apples-to-apples comparison was hard.

Minor typographical things:

Line 161: "different" has a missing character box in it.

The references in SI are jumbled in the text and missing in list form. I am having issues of my own with Word and reference software, so I can't rule it isn't my problem.

Again, this is a well put-together paper, I look forward to seeing it in print and all the future work that comes from it.

Sincerely,
Drew E. Latta

Reviewer #2 (Remarks to the Author):

Review of U Reduction Paper (Nature Comms)

The paper describes the reduction of U(VI) under anaerobic conditions with magnetite. The results from x-ray absorption methods (XAS) and electron energy loss spectroscopy (EELS) are clear, reduction has occurred. The microscopic analyses point to the formation of different and unusual microstructures that have not been observed previously owing to the different chemical conditions used and also to the lack of aberration corrected TEM/STEM analysis (according to the authors).

The reduction aspect of the work has been well studied by others and is unquestionable. The techniques used and the precautions adopted are consistent with others work. The microscopy work (EELS maps, atomic resolution images, branching ratio analysis) is all extremely good. I appreciate the authors providing high quality images for the reviewing process.

The authors state the major findings with extreme clarity in the discussion and the major question for them is that why did these nano-wire structures form and why do they fall apart and agglomerate to form other particles. The authors suggest Crystalline Particle Attachment (CPA) as a possible process – again this is not unreasonable. They finish by suggesting that cryoEM would be a perfect method for resolving this issue. I would note that cryoEM (as well as liquid cell EM) has been used by several researchers, notably Penn et al.¹, deYoreo and co-workers^{2, 3}, to examine these processes.

However, these processes have not been demonstrated in the experiments – if CPA methods were suspected, could the experiments be re-run with shorter time periods to examine these effects. This is unfortunate, as if the solution data indicated a reduction at a specific time, it would have been best to focus the microscopy effort just at this time period. If CPA was indicated or suspected, it would be best to go back and look for this effect

The first part of the paper on XAS observed reduction is completely expected but is presented in the paper as something new. Surely, we should not expect anything different – reduction of U(VI) by magnetite is well-established. However, it is new that EELS has also been used and the results from EELS are confirmed by the XAS work.

I want to turn attention to the identification of UO₂ nano-particles. The problem with determining UO₂ structures with fast Fourier transforms (FFT)s from 'nano-uraninite' is that your errors do not change just because you use an FFT. Although an FFT reflects long-range order in a crystal, when the crystal is very small, the FFT is similarly impacted. An FFT is not a selected area- or convergent beam- electron diffraction pattern, the large reduction in the total area analyzed impacts the resulting resolution of the calculated pattern. So, are the identifications that reliable? The other more important problem is that the FFT of a STEM-HAADF image is the FFT of the observable features (i.e. the uranium sub-lattice – the oxygen is too weak a scatterer to be seen by the e-beam). So it is not surprising that the structure matches UO₂. Any extra oxygens would not change the resulting pattern because you cannot see them. This is exact problem was confronted by Spurgeon et al.⁴ and it was resolved by matching multi-slice models to the observed HAADF contrast and by performing DFT analysis of oxygen O-K EELS spectra. By using this combination of methods, it was possible to show the presence of UO_{2+x} structures. Given that this type of analysis is now possible with UO₂ and other materials (btw -the authors claimed that this is the first time that aberration-corrected STEM has been used in the UO₂ system – this is not correct – the Spurgeon paper was probably the first published example of true atomic resolution imaging of UO₂) – papers that use this approach of obtaining FFTs of high resolution images (a good example is the ES&T paper by Powell and co-workers on Pu-oxide association with goethite⁵) should be reminded that this is not the same as multi-slice modeling of atomic resolution images. It is just a nice convenient method that can be done very quickly.

One note – the authors use a script to produce RDFs from the FFTs, if they are using David Mitchell's scripts they should reference his published papers on this.^{6, 7} I don't believe that GMS has an in-built scripts for this function.

The authors say that no one else has observed the nano-wires because AC-STEM has not been available; however, the observed nano-wire precipitates would be observable by conventional high-resolution microscopy – you do not need atomic resolution imaging to see 2-3 nm structures. So, why haven't they been seen before? If it is because the chemical conditions were different as the authors state then how general is this observation? Can the authors cite other microstructural analyses that show UO₂ phases forming. Should that data be re-analyzed in the light of this paper?

The idea that the nano-particles of UO₂ undergo a CPA processes is reasonable and I think this comment will influence the field to attempt this type of study, so it is a good thing to point out.

In summary

1. Some explanation of the errors associated with FFTs should be made with respect to analyzing UO₂ and that the images do not show the location of oxygens.

2. The authors have a lot of detail on the preparation of the samples for TEM. Why did they take these precautions – presumably to prevent oxidation of the U phase. Have they observed such changes. How do we know if the precautions were adequate?

References

1. Soltis, J. A.; Penn, R. L., Oriented Attachment and Nonclassical Formation in Iron Oxides. In Iron Oxides, Wiley-VCH Verlag GmbH & Co. KGaA: 2016; pp 243-268.
2. Li, D.; Nielsen, M. H.; Lee, J. R. I.; Frandsen, C.; Banfield, J. F.; De Yoreo, J. J., Direction-Specific Interactions Control Crystal Growth by Oriented Attachment. Science 2012, 336 (6084),

1014-1018.

3. De Yoreo, J. J.; Gilbert, P. U. P. A.; Sommerdijk, N. A. J. M.; Penn, R. L.; Whitlam, S.; Joester, D.; Zhang, H.; Rimer, J. D.; Navrotsky, A.; Banfield, J. F.; Wallace, A. F.; Michel, F. M.; Meldrum, F. C.; Cölfen, H.; Dove, P. M., Crystallization by particle attachment in synthetic, biogenic, and geologic environments. *Science* 2015, 349 (6247).
4. Spurgeon, S. R.; Sassi, M.; Ophus, C.; Stubbs, J. E.; Ilton, E. S.; Buck, E. C., Nanoscale oxygen defect gradients in UO_2 surfaces. *Proceedings of the National Academy of Sciences* 2019, 201905056.
5. Powell, B. A.; Dai, Z.; Zavarin, M.; Zhao, P.; Kersting, A. B., Stabilization of Plutonium Nano-Colloids by Epitaxial Distortion on Mineral Surfaces. *Environmental Science & Technology* 2011, 45 (7), 2698-2703.
6. Mitchell, D. R. G.; Petersen, T. C., RDFTools: A software tool for quantifying short-range ordering in amorphous materials. *Microscopy Research and Technique* 2012, 75 (2), 153-163.
7. Mitchell, D. R. G., DiffTools: Electron diffraction software tools for DigitalMicrograph™. *Microscopy Research and Technique* 2008, 71 (8), 588-593.

Reviewers' comments:

Reviewer #1 (Remarks to the Author):

In the present manuscript Pan and co-workers describe a new twist to an old story – reduction of U(VI) by the Fe(II,III) oxide magnetite. The new twists they put to the story are that they (i) have resolved the kinetics of the reduction of U(VI) through U(V) to U(IV) using high-resolution XAS data, and (ii) used detailed high-resolution TEM imaging and spectroscopy to further track the reduction and U(V)/U(IV) transformation process. The observation of this phased transformation from U(VI) to surface U(V) and UO₂ nanowires then to UO₂ clusters with progressively more order is very fascinating and certainly of interest to the community that studies molecular-scale mineral transformations. Detailing this step-wise U transformation process is the main finding of the work. Their findings are on par with the interesting transformations observed for Fe and Ti oxides like oriented attachment-driven crystal growth and fits well with those bodies of work. I also wonder if their data might bring into question some of the mechanisms of bacterial U reduction as well.

I was excited to review the paper when I read the abstract about the UO₂ nanowires transforming to clusters. I was always curious to know why U(IV) had formed clusters in our U(VI)/magnetite experiments at high U:magnetite loadings. The work here is the beginning of an answer to that question, along with some more interesting observations along the way.

The paper is very well written. It has to be in the top 2% of all papers I've ever reviewed for putting together the data into a concise and well-argued manuscript. Kudos to the authors on their hard work!
We thank the reviewer for his very positive assessment of our manuscript.

My recommendation is to accept the article with minor revisions. The most significant of these minor revisions would be to assess all alternative hypothesis that might explain the observations and rule them out if they can be with the data at hand – particularly the role of any solution species in the formation of the nanowires or clusters. I think an expanded section ca. line 443 detailing alternative hypotheses and why they can or cannot be rejected would be an excellent addition (or how they might need to be tested further). Some thoughts on what those alternative hypotheses could be are described below along with other specific comments.

We addressed this request by the reviewer in detail in each of the specific questions/comments below.

Other specific comments/questions:

1. So that others might repeat your experiments: what is the Fe(II)/Fe(III) stoichiometry of the magnetite used in this study? Do you have any data or left-over anoxic samples to point towards this? (But totally understood if not.) Our experience has been that washing results in variable non-stoichiometric magnetite Fe(II)/Fe(III) contents. It is likely that a moderate Fe(II)/Fe(III) ratio is the key to the slow U(VI)/U(V) reduction kinetics.

We still have the magnetite stock suspensions that were used in all the experiments (except Control 4 for which the magnetite stock was produced separately). We have now used the colorimetric method with ferrozine to determine the ratio of Fe(II)/Fe(III) and found it to be ~0.51, confirming the expected stoichiometry. We discussed in the manuscript that the magnetite reactivity might be the key to the slow U(VI)/U(V) reduction kinetics and associated production of nanowires. However, the reactivity depends on many aspects including but not limited to (a) the Fe(II)/Fe(III) ratio as Reviewer 1 suggested here; (b) the availability of Fe(II) on magnetite surface; (c) magnetite nanoparticle size; (d) aqueous chemistry conditions. And the ratio, the availability of Fe(II) and particle size all depend on synthesis processes that may differ slightly from each synthesis batch. We had already described the magnetite synthesis procedure in the method section and have now added the colorimetric method in Supplementary information. The ratio of Fe(II)/Fe(III) has now been provided in Lines 365-366, followed by an expanded discussion in Lines 366-369.

2. Are there any comparable systems in the other oxides (Fe, Ti, Si(?)) or metals (Pt, etc.) that show similar behavior as suggested here?

Nanoparticles of Fe oxides, Ti oxides, Pu oxides and other metals (Au) have been studied for their aggregation/attachment behavior during the crystal growth process in synthetic systems. In some cases, oriented attachment or coalescence occurred. Banfield et al. have reported the aggregation-based crystal growth of iron oxides in a natural sample to form ordered chains of nanoparticles and goethite rods via oriented attachment. We have now cited those studies in Lines 482-488 with more discussions on the mineral crystal growth process, which is also an important comment given by Reviewer 2 on crystallization by particle attachment (CPA).

3. Why is the assembly of the UO₂ nanowires not the (for lack of a better word) “clicking-in” type reaction observed for Fe oxides (e.g. Li et al, Science, 2012 DOI: 10.1126/science.1219643), where particles move through Brownian motion until they find a preferred low energy direction and stop and “heal” into the underlying crystal?

The 'clicking-in' of ferrihydrite nanoparticles observed by Li et al., was reported for "unattached", single nanoparticles. This is not the case here, as the UO₂ nanoparticles are bound (either to magnetite or to other UO₂ nanoparticles). However, Li et al. report that, for trapped particles unable to freely rotate, attachment to another nanoparticle resulted in attachment with alignment along different zone axes. We expect that this is what occurs in the present system and we have now added the discussion in Lines 491-497.

4. In the paragraph starting on Line 443: It is clear that the data do point towards the fact that some UO₂ nanoparticles are assembled on the surface of the magnetite. However, it is also possible that either exclusively or concurrently assemblage of the "planktonic" UO₂ particles occurs through a solution species (either U(V) or U(IV)). There is not data to rule this out, and it might be a more careful approach to include alternative hypotheses for the data in this discussion.

The reviewer is correct that we do not have direct evidence to completely discard the possibility that uraninite nanoparticles might precipitate out from solution species in the aqueous phase as we could not identify the location of the formation from static TEM images. However, based on our wet chemistry analysis, the concentration of aqueous U was less than 1% of total U after 1 hour and 0.5% after 12 hours, suggesting that the vast majority of U was associated with the solid phase (adsorption and reduction). Consequently, the dominant reduction process and the formation of U(IV) most likely occurred on the magnetite surface. We have now added the hypothesis and discussions in Lines 442-448 for clarification.

5. Along the lines of comment 4: It is less plausible to me that the assemblage of the extensive nanowire networks shown in Figure 6 occurs via the surface templating reaction over a distance of several microns. What does the rate of this reaction have to be for this to happen over 65 h? Is it plausible for the surface-templated and assembly reaction to occur over this time frame? (I don't have a good answer on whether this calculation is possible)

The conclusion that uraninite nanoparticles are formed on the magnetite surface was based on several lines of evidence. (1) Epitaxial growth was observed, and short nanowires were anchored at the magnetite surface. Thus, elongation of nanowires most likely resulted from the addition of uraninite nanoparticles to the end of nanowires, away from the magnetite surface instead of pushing formed nanowires away from the magnetite surface. (2) A negligible amount of U was measured in the solution, suggesting that the dominant reduction process and the formation of U(IV) most likely occurred on the magnetite surface instead of precipitating from the aqueous phase. However, the exact mechanism of the surface-templated and assembly reactions, thus the rate calculation, are difficult to be interpreted based on the static 2D image as it is difficult to know the original location at which the observed nanoparticles formed. As we discussed in the response to Reviewer 2's comments on crystal particle growth process, in-situ TEM experiments would be essential to understand more details and we hope the future in-situ TEM experiments are able to better answer such questions (Lines 497-499, Lines 527-530).

6. What is the solution U concentration with time in absolute concentration moles/L or ng/L? ICP-MS was used, so likely you should be able to detect and quantify down to the solubility limit of UO₂.

Yes, we have the absolute concentrations of U in the aqueous phase and we were presenting the result as the percentage of the initial total U concentration. The absolute concentration of U was 198 µg/L at 12 h and 100 µg/L at 24 h. The solubility of UO₂ is less than 10⁻² µg/L. Thus, the detected U in the aqueous phase should still be oxidized aqueous U(VI) or could represent the solubility of nanoparticulate UO₂, which may differ from that of bulk UO₂.

7. Is UO₂ conductive enough to transport electrons from magnetite to any U(VI) or U(V) at the distal end of UO₂ nanowires? E.g. is this nanowire growth an electrochemical reaction occurring at the UO₂? The nanowires remind me in a way of the dendrites you might observe for the reduction of a coinage metal at an electrode (Ag, Cu... etc.).

Before the determination of the spatial distribution of U valence states by EELS measurement, we had considered the possibility of electron transfer through UO₂ nanowires and of U(VI) reduction taking place at the end of the nanowires. However, the following data steered us away from that hypothesis: we detected U(VI) and U(V) both on the surface of magnetite nanoparticles. In contrast, overwhelmingly, U(IV) is observed in the nanowires. Additionally, less than 0.5% of U in the aqueous phase after 12 h. This suggests that most of the reduction is taking place on the magnetite surface. Otherwise, we should have observed more soluble U that would be accessible to be reduced at the distal end or adsorbed U(VI)/U(V) with nanowires.

8. What is the aqueous Fe(II) concentration? Could aqueous Fe(II) be involved in the observed behavior through aqueous U(V)/U(VI) reduction?

During the reduction experiment, we collected supernatant samples and measured total Fe concentration by ICP-OES. Total Fe measured at 1 day, 5 days, and 10 days were all around 9-12 µM, which is ~0.2% of the Fe in the solid phase. Thus, while we cannot completely exclude a role for aqueous Fe(II), we hypothesize the effect would be insignificant.

9. Why was a uranyl(V) compound not included in the analysis? I am not fully convinced that the surface U(V) is uranate(V) rather than uranyl(V). Ikeda and Hennig have published some LIII XANES and EXAFS data and the post-edge data in Figure S4 seem to fit more with the uranyl(V) shape than the post-edge of the uranate(V) compound shown (Ikeda et al 2007 Inorg. Chem. 10.1021/ic070051y). But I had trouble overlaying their data with yours crudely in powerpoint due to lower white line intensity in Ikeda, so a apples-to-apples comparison was hard.

The difference of XANES spectra features between uranyl(V) and uranate(V) mainly relies on the shape of the spectra at the white line and in post-edge features. Uranyl(V) usually shows two post-edge peaks representing the short trans-dioxo bonds and equatorial U-O bonds in the uranyl structure. Uranate(V) usually has more equivalent U-O bond distances, showing a broad white line. We have spectra for uranyl(V), and they do not correspond to the shape of the U(V) in this system. We have now clarified the differences between uranyl and uranate structures in Lines 185-192.

[REDACTED]

The broadening of the white line area was not observed in our sample (16 h sample), and the post-edge peaks in our sample spectra correspond better to those of uranyl(VI) than to those of uranyl(V).

[REDACTED]

Minor typographical things:

Line 161: "different" has a missing character box in it.

We checked Line 161 and that paragraph in the original manuscript to make sure there is no missing character box in it.

The references in SI are jumbled in the text and missing in list form. I am having issues of my own with Word and reference software, so I can't rule it isn't my problem. Again, this is a well put-together paper, I look forward to seeing it in print and all the future work that comes from it.

The references were checked and are in the proper format.

Reviewer #2 (Remarks to the Author):

The paper describes the reduction of U(VI) under anaerobic conditions with magnetite. The results from x-ray absorption methods (XAS) and electron energy loss spectroscopy (EELS) are clear, reduction has occurred. The microscopic analyses point to the formation of different and unusual microstructures that have not been observed previously owing to the different chemical conditions used and also to the lack of aberration corrected TEM/STEM analysis (according to the authors).

The reduction aspect of the work has been well studied by others and is unquestionable. The techniques used and the precautions adopted are consistent with others work. The microscopy work (EELS maps, atomic resolution images, branching ratio analysis) is all extremely good. I appreciate the authors providing high quality images for the reviewing process.

The authors state the major findings with extreme clarity in the discussion and the major question for them is that why did these nano-wire structures form and why do they fall apart and agglomerate to form other particles. The authors suggest Crystalline Particle Attachment (CPA) as a possible process – again this is not unreasonable. They finish by suggesting that cryoEM would be a perfect method for resolving this issue. I would note that cryoEM (as well as liquid cell EM) has been used by several researchers, notably Penn et al.¹, deYoreo and co-workers^{2, 3}, to examine these processes.

The reviewer's point is well taken, and we now discuss some of that work and have expanded discussions on CPA and possible future work in Lines 482-488, Lines 491-499 and Lines 527-530. We would like to clarify that the intended meaning of the original statement was: even though these tools have been applied previously, they have yet to be used to study reductive mineralization at mineral-water interfaces (particularly for U).

However, these processes have not been demonstrated in the experiments – if CPA methods were suspected, could the experiments be re-run with shorter time periods to examine these effects. This is unfortunate, as if the solution data indicated a reduction at a specific time, it would have been best to focus the microscopy effort just at this time period. If CPA was indicated or suspected, it would be best to go back and look for this effect.

We thank the Reviewer for the comments on CPA, which enabled us to examine more carefully the crystal growth process during the formation and collapse of nanowires structure. The reviewer is correct that we cannot conclusively establish the mechanisms for nanowire formation as CPA and have chosen to leave that question for further work. In-situ TEM experiments would be essential to unravel the pathways of CPA. We attempted to perform an in-situ TEM experiment with a liquid cell. However, significant beam-induced changes were observed in the preliminary test, and we chose not to report those inconclusive findings in this work. As the complexities and scope of studying CPA with in-situ liquid cell TEM go beyond the current aims of this manuscript, we prefer to leave them for future studies. It will be an independent research system, requiring a set of systematical observations on the crystallization process with the consideration of all the measurement conditions that are completely different from a batch set up. The discussion has been added in Lines 491-499 and Lines 527-530.

The first part of the paper on XAS observed reduction is completely expected but is presented in the paper as something new. Surely, we should not expect anything different – reduction of U(VI) by magnetite is well-established. However, it is new that EELS has also been used and the results from EELS are confirmed by the XAS work.

The reviewer was correct that the reduction is expected in certain aspects, such as the reduction from U(VI) to U(IV). However, the presence of U(V) was still under discussion for the reaction under neutral pH conditions, and its persistence has not been reported in such reduction reactions. U(VI) reduction to U(IV) has been well-established with L3 XAS or L3 HR XANES. For U reduction on magnetite, M4 XANES measurement has now been applied to provide strong evidence of the persistence of U(V). And the reviewer was correct that we applied EELS measurement as a new tool to localize U valence states. Furthermore, we linked the nanoscale measurement to bulk-scale measurement that allowed us to better interpret nanoscale observations given information from the bulk measurement.

I want to turn attention to the identification of UO₂ nano-particles. The problem with determining UO₂ structures with fast Fourier transforms (FFT)s from 'nano-uraninite' is that your errors do not change just because you use an FFT. Although an FFT reflects long-range order in a crystal, when the crystal is very small, the FFT is similarly impacted. *An FFT is not a selected area- or convergent beam- electron diffraction pattern, the large reduction in the total area analyzed impacts the resulting resolution of the calculated pattern.* So, are the identifications that reliable? The other more important problem is that the FFT of a STEM-HAADF image is the FFT of the observable features (i.e. the uranium sub-lattice – the oxygen is too weak a scatterer to be seen by the e-beam). So it is not surprising that the structure matches UO₂. Any extra oxygens would not change the resulting pattern because you cannot see them. This is exact problem was confronted by Spurgeon et al.⁴ and it was resolved by matching multi-slice models to the observed HAADF contrast and by performing DFT analysis of oxygen O-K EELS spectra. By using this combination of methods, it was possible to show the presence of UO_{2+x} structures. Given

that this type of analysis is now possible with UO₂ and other materials (btw -the authors claimed that this is the first time that aberration-corrected STEM has been used in the UO₂ system – this is not correct – the Spurgeon paper was probably the first published example of true atomic resolution imaging of UO₂) – papers that use this approach of obtaining FFTs of high resolution images (a good example is the ES&T paper by Powell and co-workers on Pu-oxide association with goethite⁵) should be reminded that this is not the same as multi-slice modeling of atomic resolution images. It is just a nice convenient method that can be done very quickly.

Firstly, we want to clarify that the intended meaning of the novelty of AC STEM for this system was for a reductive mineralization system. Others have used AC STEM for pre-formed UO_{2+x} as the reviewer accurately points out. We have rewritten this section in the text in Lines 372-373 and removed the section about the novelty of AC STEM as it seems to be misleading.

Regarding the use of FFT, and its limitation in terms of detecting structure oxide phases in HRSTEM, we agree and added the discussions in Supplementary Information, Lines 136-145. We have addressed this issue by performing additional measurements using selected area electron diffraction (SAED, Supplementary Information, Page 4). We probed a set of uranium oxides as references, UO₂, U₃O₈, and UO₃ as well as nanowires structures (72 hour). The results have now been discussed in the main manuscript Lines 282-286 and Supplementary Information (Fig. S7 and S8). The results clearly confirm the findings reported in the original manuscript.

One note – the authors use a script to produce RDFs from the FFTs, if they are using David Mitchell's scripts they should reference his published papers on this. 6, 7 I don't believe that GMS has an in-built scripts for this function.

Thanks to Reviewer 2 for pointing this out. We have now cited the references for the source of script we used to produce RDFs in Line 324.

The authors say that no one else has observed the nano-wires because AC-STEM has not been available; however, the observed nano-wire precipitates would be observable by conventional high-resolution microscopy – you do not need atomic resolution imaging to see 2-3 nm structures. So, why haven't they been seen before? If it is because the chemical conditions were different as the authors state then how general is this observation? Can the authors cite other microstructural analyses that show UO₂ phases forming. Should that data be re-analyzed in the light of this paper?

To determine whether such structures could have been observed in previous studies, we scoured seventeen manuscripts considering U(VI) heterogeneous reduction by Fe(II)-containing solids. Of those, only six used electron microscopy to consider the product of the reduction. Of those, only one (Latta, 2014) imaged the morphology of U precipitates as a function of time, and we hypothesize that the reduction rate was faster than here due to a lower U:Fe ratio in that study. In summary, we attribute the novel observation of UO₂ nanowires in this study to experimental factors, chiefly among them, the kinetics of U(VI) reduction and the systematic temporal characterization of the U(IV) product. We expanded our observations through one more control experiment with changed aqueous composition (at pH 6.2 and 8) to prove the presence of nanowires structure in more general conditions. The description on the control experiment has been added in method session, Lines 699-700, 714-717. The result is now been discussed in Lines 366-369, Lines 372-380 and Lines 451-454 and presented in Fig. S6e, S6f.

The idea that the nano-particles of UO₂ undergo a CPA processes is reasonable and I think this comment will influence the field to attempt this type of study, so it is a good thing to point out.

We thank the reviewer for this suggestion. We have strengthened the discussion of CPA and pointed it out as an interest for future work in Lines 527-530. The focus of the current study and the implication of uranium-magnetite system for future studies on CPA have also been strengthened and expanded in Lines 522-523 and Lines 534-538.

In summary

1. Some explanation of the errors associated with FFTs should be made with respect to analyzing UO₂ and that the images do not show the location of oxygens.

We have now included some explanation of the errors associated with FFT when analyzing UO₂ in Lines 279-280. To address Reviewer 2's concern that the FFT analysis may not contain structural information about oxygen in the lattice, we have now acquired new SAED data to further compare the crystal structure of nanowires with uranium oxides references and confirm the nanoparticles inside nanowires as UO₂ in Lines 282-286 and discussions in Supplementary Information, Fig. S7, Fig. S8.

2. The authors have a lot of detail on the preparation of the samples for TEM. Why did they take these precautions – presumably to prevent oxidation of the U phase. Have they observed such changes. How do we know if the precautions were adequate?

The precautions were to prevent the oxidation of samples during the preparation of TEM grids as the possibility of oxidation of U(IV)/U(V) to U(VI). Though we did not observe any changes, we implemented appropriate experimental practices and rigor and minimized the contact of samples with oxygen as much as possible. During the measurement, at least there was no oxidation of the nanowires because now EELS, FFT, and SAED all confirm that they are U(IV). And the presence of U(V) confirmed by EELS is

consistent with the interpretation of valence states from XAS measurement. However, during the measurement, beam-induced reduction was a concern that we tried to avoid as much as possible. We did notice that one sample changed under the beam (UO₃ standard sample), and we provided those details to help guide other researchers in the future. Changes were observed in the UO₃ sample for the EELS measurement and the newly performed SAED, both of which are now discussed in the Supplementary information, Lines 131-135, Lines 169-173.

References

1. Soltis, J. A.; Penn, R. L., Oriented Attachment and Nonclassical Formation in Iron Oxides. In Iron Oxides, Wiley-VCH Verlag GmbH & Co. KGaA: 2016; pp 243-268.
2. Li, D.; Nielsen, M. H.; Lee, J. R. I.; Frandsen, C.; Banfield, J. F.; De Yoreo, J. J., Direction-Specific Interactions Control Crystal Growth by Oriented Attachment. *Science* 2012, 336 (6084), 1014-1018.
3. De Yoreo, J. J.; Gilbert, P. U. P. A.; Sommerdijk, N. A. J. M.; Penn, R. L.; Whitlam, S.; Joester, D.; Zhang, H.; Rimer, J. D.; Navrotsky, A.; Banfield, J. F.; Wallace, A. F.; Michel, F. M.; Meldrum, F. C.; Cölfen, H.; Dove, P. M., Crystallization by particle attachment in synthetic, biogenic, and geologic environments. *Science* 2015, 349 (6247).
4. Spurgeon, S. R.; Sassi, M.; Ophus, C.; Stubbs, J. E.; Ilton, E. S.; Buck, E. C., Nanoscale oxygen defect gradients in UO_{2+x} surfaces. *Proceedings of the National Academy of Sciences* 2019, 201905056.
5. Powell, B. A.; Dai, Z.; Zavarin, M.; Zhao, P.; Kersting, A. B., Stabilization of Plutonium Nano-Colloids by Epitaxial Distortion on Mineral Surfaces. *Environmental Science & Technology* 2011, 45 (7), 2698-2703.
6. Mitchell, D. R. G.; Petersen, T. C., RDTTools: A software tool for quantifying short-range ordering in amorphous materials. *Microscopy Research and Technique* 2012, 75 (2), 153-163.
7. Mitchell, D. R. G., DiffTools: Electron diffraction software tools for DigitalMicrograph™. *Microscopy Research and Technique* 2008, 71 (8), 588-593.

We have now cited the references for the source of script we used to produce RDFs and discussed those references on CPA.

REVIEWERS' COMMENTS:

Reviewer #1 (Remarks to the Author):

I have read the revised manuscript and responses to my review comments by Pan et al. I believe that the authors have sufficiently answered these questions and have provided appropriate edits, where necessary to improve the manuscript. At this time I have no further comments. The manuscript could be accepted as-is at this time.

Best regards,

Drew E. Latta

=====

Drew E. Latta, Ph.D.

Asst. Research Scientist

The University of Iowa
IIHR/Civil and Environmental Engineering
4105 Seamans Center
Iowa City, IA 52242

=====

Reviewer #2 (Remarks to the Author):

The authors have addressed all the comments.

The reviewers did not request any additional changes to our manuscript. We appreciate the reviewers' constructive comments and suggestions, as well as the positive assessment on our work.

REVIEWERS' COMMENTS:

Reviewer #1 (Remarks to the Author):

I have read the revised manuscript and responses to my review comments by Pan et al. I believe that the authors have sufficiently answered these questions and have provided appropriate edits, where necessary to improve the manuscript. At this time, I have no further comments. The manuscript could be accepted as-is at this time.

Best regards,
Drew E. Latta

Drew E. Latta, Ph.D.
Asst. Research Scientist
The University of Iowa
IIHR/Civil and Environmental Engineering
4105 Seamans Center
Iowa City, IA 52242

Reviewer #2 (Remarks to the Author):

The authors have addressed all the comments.